# Epsilon-VAE: Denoising as Visual Decoding

**Long Zhao** [1]  **Sanghyun Woo** [1]  **Ziyu Wan** [1 2 *]  **Yandong Li** [1]  **Han Zhang** [1]  **Boqing Gong** [1]  **Hartwig Adam** [1]
**Xuhui Jia** [1]  **Ting Liu** [1]

## Abstract

In generative modeling, tokenization simplifies complex data into compact, structured representations, creating a more efficient, learnable space. For high-dimensional visual data, it reduces redundancy and emphasizes key features for high-quality generation. Current visual tokenization methods rely on a traditional autoencoder framework, where the encoder compresses data into latent representations, and the decoder reconstructs the original input. In this work, we offer a new perspective by proposing *denoising as decoding*, shifting from single-step reconstruction to iterative refinement. Specifically, we replace the decoder with a diffusion process that iteratively refines noise to recover the original image, guided by the latents provided by the encoder. We evaluate our approach by assessing both reconstruction (rFID) and generation quality (FID), comparing it to state-of-the-art autoencoding approaches. By adopting iterative reconstruction through diffusion, our autoencoder, namely $\epsilon$-VAE, achieves high reconstruction quality, which in turn enhances downstream generation quality by 22% at the same compression rates or provides $2.3\times$ inference speedup through increasing compression rates. We hope this work offers new insights into integrating iterative generation and autoencoding for improved compression and generation.

## 1. Introduction

Two dominant paradigms in modern visual generative modeling are autoregression (Radford et al., 2018) and diffusion (Ho et al., 2020). Tokenization is essential for both: discrete tokens allow step-by-step conditional generation in autoregressive models, while continuous latents enable efficient learning in the denoising process of diffusion models. In either case, empirical results demonstrate that tokenization enhances generative performance. Here, we focus on *continuous* tokenization for latent diffusion models, which excel at generating high-dimensional visual data.

In this paper, we revisit the conventional autoencoding pipeline, which typically consists of an encoder that compresses the input into a latent representation and a decoder that reconstructs the original data in a single step. Instead of a deterministic decoder, we introduce a diffusion process (Ho et al., 2020; Song et al., 2021), where the encoder still compresses the input into a latent representation, but reconstruction is performed iteratively through denoising. This reframing turns the reconstruction phase into a progressive refinement process, where the diffusion model, guided by the latent representation, gradually restores the original data. While previous work (Preechakul et al., 2022) and concurrent work (Birodkar et al., 2024) have explored diffusion mechanisms in autoencoding, none have fully realized a practical diffusion-based autoencoder. By carefully co-designing architecture and objectives, we firstly show that our approach outperforms state-of-the-art autoencoding paradigms in reconstruction fidelity, sampling efficiency, and resolution generalization.

To effectively implement our approach, several key design factors must be carefully considered. First, the architectural design must ensure that the diffusion decoder is effectively conditioned on the encoder latent representations. Second, the training objectives should leverage synergies with traditional autoencoding losses, such as LPIPS (Zhang et al., 2018) and GAN (Esser et al., 2021). Finally, diffusion-specific design choices play a crucial role, including: (1) model parameterization, which defines the prediction target for the diffusion decoder; (2) noise scheduling, which shapes the optimization trajectory; and (3) the distribution of time steps during training and testing, which balances noise levels for effective learning and generation. Our study systematically examines these components through controlled experiments, demonstrating their impact on achieving a high-performing diffusion-based autoencoder. We show in the experiments that under the standard configuration (Rombach et al., 2022), our method obtains a 40% improvement in

---

*Work done as a student researcher at Google. [1]Google DeepMind [2]City University of Hong Kong. Correspondence to: Long Zhao <longzh@google.com>, Xuhui Jia <xhjia@google.com>, Ting Liu <liuti@google.com>.

terms of reconstruction quality, leading to 22% better image generation quality. More notably, we achieve $2.3\times$ higher inference throughput by increasing compression rates, while keeping competitive generation quality.

In summary, our contributions are as follows: (1) introducing a novel approach that fully leverages the capabilities of diffusion decoders for more practical diffusion-based autoencoding, achieving strong rFID, high sampling efficiency (within 1 to 3 steps), and robust resolution generalization; (2) presenting key design choices in both architecture and objectives to optimize performance; and (3) conducting extensive controlled experiments that demonstrate our method achieves high-quality reconstruction and generation results, outperforming leading visual auto-encoding paradigms.

## 2. Background

We start by briefly reviewing the basic concepts required to understand the proposed method. A more detailed summary of related work is deferred to Appx. B.

**Visual autoencoding.** To achieve efficient and scalable high-resolution image synthesis, common generative models, including autoregressive models (Razavi et al., 2019; Esser et al., 2021; Chang et al., 2022) and diffusion models (Rombach et al., 2022), are typically trained in a low-resolution latent space by first downsampling the input image using a tokenizer. The tokenizer is generally implemented as a convolutional autoencoder consisting of an encoder, $\mathcal{E}$, and a decoder, $\mathcal{G}$. Specifically, the encoder, $\mathcal{E}$, compresses an input image $\boldsymbol{x} \in \mathbb{R}^{H \times W \times 3}$ into a set of latent codes (*i.e.*, tokens), $\mathcal{E}(\boldsymbol{x}) = \boldsymbol{z} \in \mathbb{R}^{H/f \times W/f \times n_z}$, where $f$ is the downsampling factor and $n_z$ is the latent channel dimensions. The decoder, $\mathcal{G}$, then reconstructs the input from $\boldsymbol{z}$, such that $\mathcal{G}(\boldsymbol{z}) = \boldsymbol{x}$.

Training an autoencoder primarily involves several losses: reconstruction loss $\mathcal{L}_{\text{rec}}$, perceptual loss (LPIPS) $\mathcal{L}_{\text{LPIPS}}$, and adversarial loss $\mathcal{L}_{\text{adv}}$. The reconstruction loss minimizes pixel differences (*i.e.*, typically measured by the $\ell_1$ or $\ell_2$ distance) between $\boldsymbol{x}$ and $\mathcal{G}(\boldsymbol{z})$. The LPIPS loss (Zhang et al., 2018) enforces high-level structural similarities between inputs and reconstructions by minimizing differences in their intermediate features extracted from a pre-trained VGG network (Simonyan & Zisserman, 2015). The adversarial loss (Esser et al., 2021) introduces a discriminator, $\mathcal{D}$, which encourages more photorealistic outputs by distinguishing between real images, $\mathcal{D}(\boldsymbol{x})$, and reconstructions, $\mathcal{D}(\mathcal{G}(\boldsymbol{z}))$. The final training objective is a weighted combination of these losses:

$$\mathcal{L}_{\text{VAE}} = \mathcal{L}_{\text{rec}} + \lambda_{\text{LPIPS}} \cdot \mathcal{L}_{\text{LPIPS}} + \lambda_{\text{adv}} \cdot \mathcal{L}_{\text{adv}}, \quad (1)$$

where the $\lambda$ values are weighting coefficients. In this paper, we consider the autoencoder optimized by Eq. 1 as our main

competing baseline (Esser et al., 2021), as it has become a standard tokenizer training scheme widely adopted in state-of-the-art image and video generative models (Chang et al., 2022; Rombach et al., 2022; Yu et al., 2022; 2023; Kondratyuk et al., 2024; Esser et al., 2024).

**Diffusion.** Given a data distribution $p_{\boldsymbol{x}}$ and a noise distribution $p_{\boldsymbol{\epsilon}}$, a diffusion process progressively corrupts clean data $\boldsymbol{x}_0 \sim p_{\boldsymbol{x}}$ by adding noise $\boldsymbol{\epsilon} \sim p_{\boldsymbol{\epsilon}}$ and then reverses this corruption to recover the original data (Song & Ermon, 2019; Ho et al., 2020), represented as:

$$\boldsymbol{x}_t = \alpha_t \cdot \boldsymbol{x}_0 + \sigma_t \cdot \boldsymbol{\epsilon}, \quad (2)$$

where $t \in [0, \text{T}]$ and $\boldsymbol{\epsilon}$ is drawn from a standard Gaussian distribution, $p_{\boldsymbol{\epsilon}} = \mathcal{N}(0, I)$. The functions $\alpha_t$ and $\sigma_t$ govern the trajectory between clean data and noise, affecting both training and sampling. The basic parameterization in Ho et al. (2020) defines $\sigma_t = \sqrt{1 - \alpha_t^2}$ with $\alpha_t = \left(\prod_{s=0}^{t}(1 - \beta_s)\right)^{\frac{1}{2}}$ for discrete timesteps. The diffusion coefficients $\beta_t$ are linearly interpolated values between $\beta_0$ and $\beta_{T-1}$ as $\beta_t = \beta_0 + \frac{t}{T-1}(\beta_{T-1} - \beta_0)$, with start and end values are set empirically.

The forward and reverse diffusion processes are described by the following factorizations:

$$q(\boldsymbol{x}_{\Delta t:\text{T}}|\boldsymbol{x}_0) = \prod_{i=1}^{\text{T}} q(\boldsymbol{x}_{i \cdot \Delta t}|\boldsymbol{x}_{(i-1) \cdot \Delta t})$$
$$\text{and } p(\boldsymbol{x}_{0:\text{T}}) = p(\boldsymbol{x}_{\text{T}}) \prod_{i=1}^{\text{T}} p(\boldsymbol{x}_{(i-1) \cdot \Delta t}|\boldsymbol{x}_{i \cdot \Delta t}), \quad (3)$$

where the forward process $q(\boldsymbol{x}_{\Delta t:\text{T}}|\boldsymbol{x}_0)$ transitions clean data $\boldsymbol{x}_0$ to noise $\boldsymbol{x}_{\text{T}} = \boldsymbol{\epsilon}$, while the reverse process $p(\boldsymbol{x}_{0:\text{T}})$ recovers clean data from noise. $\Delta t$ denotes the time step interval or step size.

During training, the model learns the score function $\nabla \log p_t(\boldsymbol{x}) \propto -\frac{\boldsymbol{\epsilon}}{\sigma_t}$, which represents gradient pointing toward the data distribution along the noise-to-data trajectory. In practice, the model $s_\Theta(\boldsymbol{x}_t, t)$ is optimized by minimizing the score-matching objective:

$$\mathcal{L}_{\text{score}} = \min_{\Theta} \mathbb{E}_{t \sim \pi(t), \boldsymbol{\epsilon} \sim \mathcal{N}(0, I)} \left[ w_t \|\sigma_t s_\Theta(\boldsymbol{x}_t, t) + \boldsymbol{\epsilon}\|^2 \right], \quad (4)$$

where $\pi(t)$ defines the time-step sampling distribution and $w_t$ is a time-dependent weight. These elements together influence which time steps or noise levels are prioritized during training. Conceptually, the diffusion model learns the tangent of the trajectory at each point along the path. During sampling, it progressively recovers clean data from noise based on its predictions.

**Rectified flow** provides a specific parametrization of $\alpha_t$ and $\sigma_t$ such that the trajectory between data and noise follows a

"straight" path (Liu et al., 2023; Albergo & Vanden-Eijnden, 2023). This trajectory is represented as:

$$\boldsymbol{x}_t = (1 - t) \cdot \boldsymbol{x}_0 + t \cdot \boldsymbol{\epsilon}, \tag{5}$$

where $t \in [0, 1]$. In this formulation, the gradient along the trajectory, $\boldsymbol{\epsilon} - \boldsymbol{x}_0$, is deterministic, often referred to as the velocity. The model $v_\Theta(\boldsymbol{x}_t, t)$ is parameterized to predict velocity by minimizing:

$$\min_\Theta \mathbb{E}_{t \sim \pi(t), \boldsymbol{\epsilon} \sim \mathcal{N}(0, I)} \left[ \| v_\Theta(\boldsymbol{x}_t, t) - (\boldsymbol{\epsilon} - \boldsymbol{x}) \|^2 \right]. \tag{6}$$

We note that this objective is equivalent to a score matching form (Eq. 4), with the weight $w_t = \left( \frac{1}{1-t} \right)^2$. This equivalence highlights that alternative model parameterizations reduce to a standard denoising objective, where the primary difference lies in the time-dependent weighting functions and the corresponding optimization trajectory (Kingma & Gao, 2024).

During sampling, the model follows a simple probability flow ODE:

$$\mathrm{d}\boldsymbol{x}_t = v_\Theta(\boldsymbol{x}_t, t) \cdot \mathrm{d}t. \tag{7}$$

Although a perfect straight path could theoretically be solved in a single step, the independent coupling between data and noise often results in curved trajectories, necessitating multiple steps to generate high-quality samples (Liu et al., 2023; Lee et al., 2024). In practice, we iteratively apply the standard Euler solver (Euler, 1845) to sample data from noise.

## 3. Method

We introduce $\epsilon$-VAE, with an overview provided in Fig. 1. The core idea is to replace single-step, deterministic decoding with an iterative, stochastic denoising process. By reframing autoencoding as a conditional denoising problem, we anticipate two key improvements: (1) more effective generation of latent representations, allowing the downstream latent diffusion model to learn more efficiently, and (2) enhanced decoding quality due to the iterative and stochastic nature of the diffusion process.

We systematically explore the design space of model architecture, objectives, and diffusion training configurations, including noise and time scheduling. While this work primarily focuses on generating continuous latents for latent diffusion models, the concept of iterative decoding could also be extended to discrete tokens, which we leave for future exploration.

### 3.1. Modeling

$\epsilon$-VAE retains the encoder $\mathcal{E}$ while enhancing the decoder $\mathcal{G}$ by incorporating a diffusion model, transforming the standard decoding process into an iterative denoising task.

**Conditional denoising.** Specifically, the input $\boldsymbol{x} \sim p_{\boldsymbol{x}}$ is encoded by the encoder as $\boldsymbol{z} = \mathcal{E}(\boldsymbol{x})$, and this encoding serves as a condition to guide the subsequent denoising process. This reformulates the reverse process in Eq. 3 into a conditional form (Nichol & Dhariwal, 2021):

$$p(\boldsymbol{x}_{0:\mathrm{T}}|\boldsymbol{z}) = p(\boldsymbol{x}_\mathrm{T}) \prod_{i=1}^{\mathrm{T}} p(\boldsymbol{x}_{(i-1) \cdot \Delta t}|\boldsymbol{x}_{i \cdot \Delta t}, \boldsymbol{z}), \tag{8}$$

where the denoising process from the noise $\boldsymbol{x}_\mathrm{T} = \boldsymbol{\epsilon}$ to the input $\boldsymbol{x}_0 = \boldsymbol{x}$, is additionally conditioned on $\boldsymbol{z}$ over time. Here, the decoder is no longer deterministic, as the process starts from random noise. For a more detailed discussion on this autoencoding formulation, we refer readers to Appx. A.

**Architecture and conditioning.** We adopt the standard U-Net architecture from Dhariwal & Nichol (2021) for our diffusion decoder $\mathcal{G}$, while also exploring Transformer-based models (Peebles & Xie, 2023). For conditional denoising, we concatenate the conditioning signal with the input channel-wise, following the approach of diffusion-based super-resolution models (Ho et al., 2022; Saharia et al., 2022b). Specifically, low-resolution latents are upsampled using nearest-neighbor interpolation to match the resolution of $\boldsymbol{x}_t$, then concatenated along the channel dimension. In Appx. D, although we experimented with conditioning via AdaGN (Nichol & Dhariwal, 2021), it did not yield significant improvement and introduced additional overhead, so we adopt channel concatenation.

### 3.2. Objectives

We adopt the standard autoencoding objective from Eq. 1 to train $\epsilon$-VAE, with a key modification: replacing the reconstruction loss $\mathcal{L}_{\text{rec}}$ used for the standard decoder with the score-matching loss $\mathcal{L}_{\text{score}}$ for training the diffusion decoder. Additionally, we introduce a strategy to adjust the perceptual $\mathcal{L}_{\text{LPIPS}}$ and adversarial $\mathcal{L}_{\text{adv}}$ losses to better align with the diffusion decoder training.

**Velocity prediction.** We adopt the rectified flow parameterization, utilizing a linear optimization trajectory between data and noise, combined with velocity-matching objective (Eq. 6). We inject the encoder output $\boldsymbol{z}$ into the objective by replacing $v_\Theta(\boldsymbol{x}_t, t)$ with $\mathcal{G}(\boldsymbol{x}_t, t, \boldsymbol{z})$.

**Perceptual matching.** The LPIPS loss (Zhang et al., 2018) minimizes the perceptual distance between the reconstructions and real images using pre-trained models, typically VGG network (Esser et al., 2021; Yu et al., 2023; 2022). We apply this feature-matching objective to train $\epsilon$-VAE. However, unlike traditional autoencoders, $\epsilon$-VAE predicts velocity instead of directly reconstructing the image during training, making it infeasible to compute the LPIPS loss directly between the prediction and the target image. To address this, we leverage the simple reversing step from

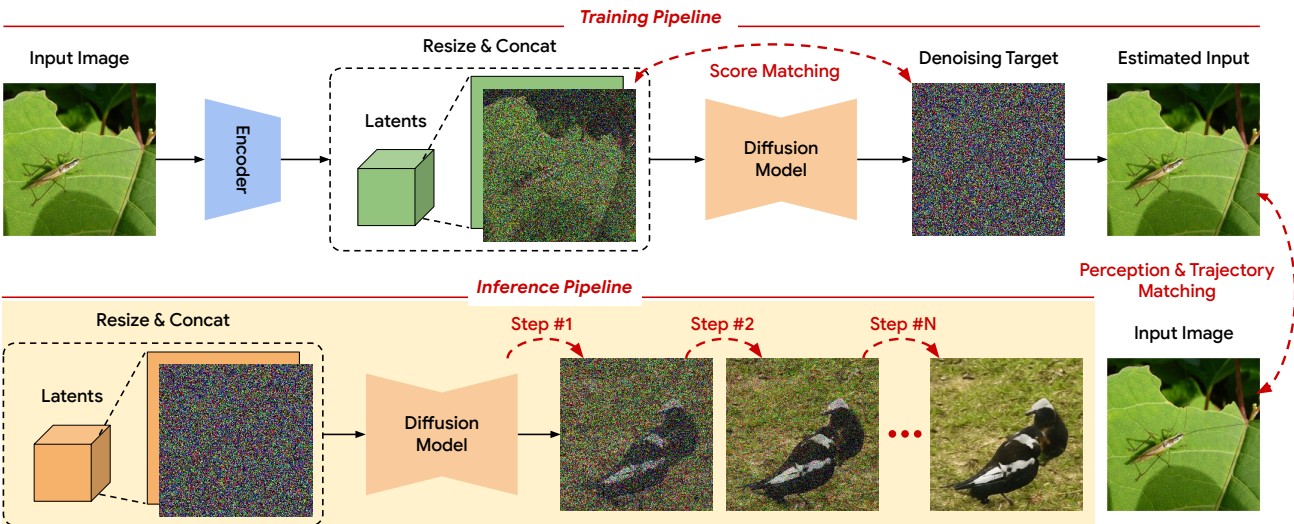

*Figure 1.* **An overview of $\epsilon$-VAE**. We frame visual decoding as an iterative denoising problem by replacing the autoencoder decoder with a diffusion model, optimized using a combination of score, perception, and trajectory matching losses. During inference, images are reconstructed (or generated) from encoded (or sampled) latents through an iterative denoising process. The number of sampling steps $N$ can be flexibly adjusted within small NFE regimes (from 1 to 3). We empirically confirm that $\epsilon$-VAE significantly outperforms the standard VAE schema, even with just a few steps.

Eq. 6 to estimate $\boldsymbol{x}_0$ from the prediction and $\boldsymbol{x}_t$ as follows:

$$\hat{\boldsymbol{x}}_0^t = \boldsymbol{x}_t - t \cdot \mathcal{G}(\boldsymbol{x}_t, t, \boldsymbol{z}), \tag{9}$$

where $\hat{\boldsymbol{x}}_0^t$ represents the reconstructed image estimated by the model at time $t$. We then compute the LPIPS loss between $\hat{\boldsymbol{x}}_0^t$ and the target real image $\boldsymbol{x}$.

**Denoising trajectory matching.** The adversarial loss encourages photorealistic outputs by comparing the reconstructions to real images. We modify this to better align with a diffusion decoder. Specifically, our approach adapts the standard adversarial loss to enforce trajectory consistency rather than solely on realism. In practice, we achieve this by minimizing the following divergence, $\mathcal{D}_{\text{adv}}$:

$$\min_{\Theta} \mathbb{E}_{t \sim p_t} \left[ \mathcal{D}_{\text{adv}} \left( q(\boldsymbol{x}_0 | \boldsymbol{x}_t) || p_{\Theta}(\hat{\boldsymbol{x}}_0^t | \boldsymbol{x}_t) \right) \right], \tag{10}$$

where $\mathcal{D}_{\text{adv}}$ is a probability distance metric (Goodfellow et al., 2014; Arjovsky et al., 2017), and we adopt the basic non-saturating GAN (Goodfellow et al., 2014).

For adversarial training, we design a time-dependent discriminator that takes time as input using AdaGN approach (Dhariwal & Nichol, 2021). To simulate the trajectory, we concatenate $\boldsymbol{x}_0$ and $\boldsymbol{x}_t$ along the channel dimension. The generator parameterized by $\Theta$, and the discriminator, parameterized by $\Phi$, are then optimized through a minimax game as:

$$\min_{\Theta} \max_{\Phi} \mathcal{L}_{\text{adv}} = \mathbb{E}_{q(\boldsymbol{x}_0 | \boldsymbol{x}_t)} [\log \mathcal{D}_{\Phi}(\boldsymbol{x}_0, \boldsymbol{x}_t, t)]$$
$$+ \mathbb{E}_{p_{\Theta}(\hat{\boldsymbol{x}}_0^t | \boldsymbol{x}_t)} \left[ \log \left( 1 - \mathcal{D}_{\Phi}(\hat{\boldsymbol{x}}_0^t, \boldsymbol{x}_t, t) \right) \right], \tag{11}$$

where fake trajectories $p_{\Theta}(\hat{\boldsymbol{x}}_0^t | \boldsymbol{x}_t)$ are contrasted with real trajectories $q(\boldsymbol{x}_0 | \boldsymbol{x}_t)$. To further stabilize training, we apply the $R_1$ gradient penalty to the discriminator parameters (Mescheder et al., 2018). In Appx. D, we explore alternative matching approaches, including the standard adversarial method of comparing individual reconstructions $\hat{\boldsymbol{x}}_0^t$ with real images $\boldsymbol{x}_0$, matching the trajectory steps $\boldsymbol{x}_t \rightarrow \boldsymbol{x}_{t-\Delta t}$ (Xiao et al., 2022; Wang et al., 2024a), and our start-to-end trajectory matching $\boldsymbol{x}_t \rightarrow \boldsymbol{x}_0$, with the latter showing the best performance.

**Final training objective** combines $\mathcal{L}_{\text{score}}$, $\mathcal{L}_{\text{LPIPS}}$, and $\mathcal{L}_{\text{adv}}$, with empirically adjusted weights (see Appx. C.2).

Note that applying LPIPS and adversarial losses on the estimated one-step sample could lead to potential objective bias. However, we would like to emphasize that $\epsilon$-VAE differs significantly from traditional diffusion models in that its diffusion decoder is conditioned on encoded latents $\boldsymbol{z}$. This conditioning provides a strong prior about the input image to reconstruct, resulting in a more accurate estimated one-step sample than in typical diffusion scenarios. Therefore, we believe the potential for objective bias is considerably reduced in $\epsilon$-VAE. Fine-tuning the diffusion decoder with frozen $\boldsymbol{z}$ like Sargent et al. (2025) could be a promising avenue for further improvement, which we will explore in our future work.

### 3.3. Noise and time scheduling

**Noise scheduling.** In diffusion models, noise scheduling involves progressively adding noise to the data over time

by defining specific functions for $\alpha_t$ and $\sigma_t$ in Eq. 2. This process is crucial as it determines the signal-to-noise ratio, $\lambda_t = \frac{\alpha_t^2}{\sigma_t^2}$, which directly influences training dynamics. Noise scheduling can also be adjusted by scaling the intermediate states $\boldsymbol{x}_t$ with a constant factor $\gamma \in (0, 1]$, which shifts the signal-to-noise ratio downward. This makes training more challenging over time while preserving the shape of the trajectory (Chen, 2023).

In this work, we define $\alpha_t$ and $\sigma_t$ according to rectified flow formulation, while also scaling $\boldsymbol{x}_t$ by $\gamma$, with the value chosen empirically. However, when $\gamma \neq 1$, the variance of $\boldsymbol{x}_t$ changes, which can degrade performance (Karras et al., 2022). To address this, we normalize the denoising input $\boldsymbol{x}_t$ by its variance after scaling, ensuring it preserves unit variance over time (Chen, 2023).

**Time scheduling.** Another important aspect in diffusion models is time scheduling for both training and sampling, controlled by $\pi(t)$ during training and $\Delta t$ during sampling, as outlined in Eq. 3 and Eq. 4. A common choice for $\pi(t)$ is the uniform distribution $\mathcal{U}(0, T)$, which applies equal weight to each time step during training. Similarly, uniform time steps $\Delta t = \frac{1}{T}$ are typically used for sampling. However, to improve model performance on more challenging time steps and focus on noisy regimes during sampling, the time scheduling strategy should be adjusted accordingly.

In this work, we sample $t$ from a logit-normal distribution (Atchison & Shen, 1980), which emphasizes intermediate timesteps (Esser et al., 2024). During sampling, we apply a reversed logarithm mapping $\rho_{\log}$ defined as:

$$\rho_{\log}(t; m, n) = \frac{\log(m) - \log\left(t \cdot (m - n) + n\right)}{\log(m) - \log(n)}, \quad (12)$$

where we set $m = 1$ and $n = 100$, resulting in denser sampling steps early in the inference process.

# 4. Experiments

We evaluate the effectiveness of $\epsilon$-VAE on image reconstruction and generation tasks using the ImageNet (Deng et al., 2009). The VAE formulation by Esser et al. (2021) serves as a strong baseline due to its widespread use in modern image generative models (Rombach et al., 2022; Peebles & Xie, 2023; Esser et al., 2024). We perform controlled experiments to compare reconstruction and generation quality by varying model scale, latent dimension, downsampling rates, and input resolution.

**Model configurations.** We use the encoder and discriminator architectures from VQGAN (Esser et al., 2021) and keep consistent across all models. The decoder design follows BigGAN (Brock et al., 2019) for VAE and from ADM (Dhariwal & Nichol, 2021) for $\epsilon$-VAE. Additionally, we experiment with the DiT architecture (Peebles & Xie,

2023) for $\epsilon$-VAE. To evaluate model scaling, we test five decoder variants: base (B), medium (M), large (L), extra-large (XL), and huge (H), by adjusting width and depth accordingly. Further model specifications are provided in Appx. C.1.

We experiment with the following two encoder configurations. $\epsilon$**-VAE-lite**: a light-weight version with 6M parameters, a downsampling rate of 16, and 8 latent channels; $\epsilon$**-VAE-SD**: a standard version based on Stable Diffusion with 34M parameters, a downsampling rate of 8, and 4 latent channels. $\epsilon$-VAE-lite is intentionally designed as a more challenging setup and serves as the primary focus of analysis in the paper. For this configuration, we further explore downsampling rates of 4, 8, and 32, as well as latent dimensions of 4, 16, and 32 channels. Both VAE and $\epsilon$-VAE are trained to reconstruct $128 \times 128$ images under these controlled conditions. Additionally, we validate our method in the standard setup of $\epsilon$-VAE-SD, where we compare it against state-of-the-art VAEs.

**Evaluation.** We evaluate the autoencoder on both reconstruction and generation quality using Fréchet Inception Distance (FID) (Heusel et al., 2017) as the primary metric, and we also report PSNR and SSIM as secondary metrics. For reconstruction quality (rFID), FID is computed at both training and higher resolutions to assess generalization across resolutions. For generation quality (FID), we generate latents from the trained autoencoders and use them to train the DiT-XL/2 latent generative model (Peebles & Xie, 2023). This latent model remains fixed across all generation experiments, meaning improved autoencoder latents directly enhance generation quality.

## 4.1. Reconstruction quality

**Decoder architecture.** We explore two major architectural designs: the UNet-based architecture from ADM (Dhariwal & Nichol, 2021) and the Transformer-based DiT (Peebles & Xie, 2023). We compare various model sizes–ADM:{B, M, L, XL, H} and DiT:{S, B, L, XL} with patch sizes of {4, 8}. The results are summarized in Fig. 2 (left). ADM consistently outperforms DiT across the board. While we observe rFID improvements in DiT when increasing the number of tokens by reducing patch size, this comes with significant computational overhead. The overall result aligns with the original design intentions: ADM for pixel-level generation and DiT for latent-level generation. For the following experiments, we use the ADM architecture for our diffusion decoder.

**Compression rate.** Compression can be achieved by adjusting either the channel dimensions of the latents or the downsampling factor of the encoder. In Fig. 2 (middle and right), we compare VAE and $\epsilon$-VAE across these two aspects. The results show that $\epsilon$-VAE consistently outperforms VAE

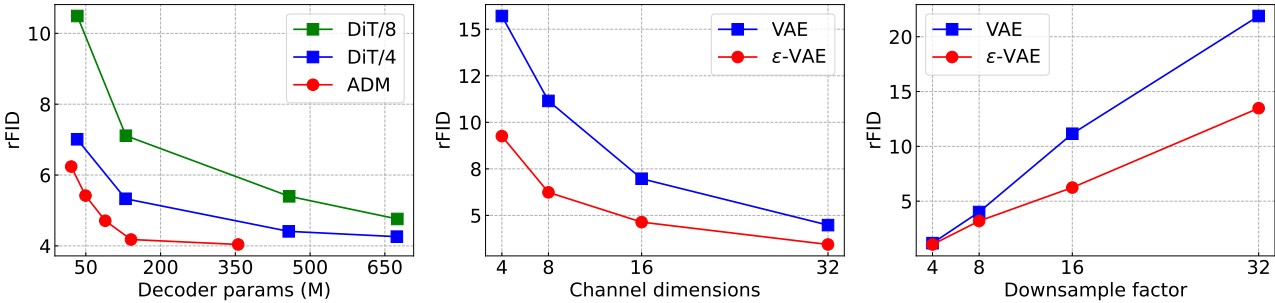

*Figure 2.* **Architecture and compression analysis.** The $\epsilon$-VAE decoder uses either a UNet-based ADM or Transformer-based DiT (left). $\epsilon$-VAE and VAE are evaluated under different compression rates by varying latent channel dimensions (middle) or encoder downsampling factors (right). *We follow the $\epsilon$-VAE-lite configuration in these experiments.*

*Table 1.* **ImageNet reconstruction results (rFID) at different resolutions using VAEs trained at** $128 \times 128$. † denotes training at $128 \times 128$ followed by fine-tuning at a higher resolution.

| Resolution (ImageNet) | 128 | 256 | 512 | 256† |
|---|---|---|---|---|
| SD-VAE (Rombach et al., 2022) | 4.54 | 1.21 | 0.91 | 0.86 |
| LiteVAE (Sadat et al., 2024) | 4.40 | 0.97 | - | 0.73 |
| $\epsilon$-VAE-SD (B) | 1.94 | 0.65 | 0.61 | 0.52 |
| $\epsilon$-VAE-SD (M) | 1.58 | 0.55 | 0.53 | 0.47 |
| $\epsilon$-VAE-SD (L) | 1.47 | 0.52 | 0.41 | 0.45 |
| $\epsilon$-VAE-SD (XL) | 1.34 | 0.49 | 0.39 | 0.43 |
| $\epsilon$-VAE-SD (H) | 1.00 | 0.44 | 0.35 | 0.38 |

in terms of rFID, particularly as the compression ratio increases. Specifically, as shown on the middle graph, $\epsilon$-VAE achieves lower rFIDs than VAE across all channel dimensions, with a notable gap at lower dimensions (4 and 8). On the right graph, $\epsilon$-VAE maintains lower rFIDs than VAE even as the downsampling factor increases, with the gap widening significantly at larger factors (16 and 32). Furthermore, $\epsilon$-VAE delivers comparable or superior rFIDs even when the compression ratio is doubled, demonstrating its robustness and effectiveness in high-compression scenarios.

**Resolution generalization.** A notable feature of conventional autoencoders is their capacity to generalize and reconstruct images at higher resolutions during inference (Rombach et al., 2022). To assess this, we conduct inference on images with resolutions of $256 \times 256$ and $512 \times 512$, using $\epsilon$-VAE and VAE models trained at $128 \times 128$. As shown in Tab. 1, $\epsilon$-VAE effectively generalizes to higher resolutions, consistently preserving its performance advantage over other VAEs. Furthermore, we find that fine-tuning models at the target (higher) resolution leads to improvement at it, which is consistent with the observation made by Sadat et al. (2024). We hence utilize this multi-stage training strategy in the following experiments when the target resolution is larger than $128 \times 128$.

**Comparisons to state-of-the-art VAEs.** We provide image reconstruction results under the same configuration as VAEs in Stable Diffusion (SD-VAE): an encoder with 34M parameters and a channel dimension of 4 for $256 \times 256$ image reconstruction. We evaluate rFID, PSNR and SSIM on the full validation sets of ImageNet and COCO-2017 (Lin et al., 2014), with the results summarized in Tab. 2. Our finds reveal that $\epsilon$-VAE outperforms state-of-the-art VAEs when the decoder sizes are comparable, and its performance can be further improved by scaling up the decoder. This demonstrates the strong model scalability of our framework.

**One-step $\epsilon$-VAE.** Note that the denoising process of $\epsilon$-VAE demonstrates promising results even with a single iteration. To show this, we provide a direct comparison between SD-VAE and our one-step $\epsilon$-VAE models in Tab. 3. This table presents image reconstruction quality on ImageNet $256 \times 256$ with the $8 \times 8$ downsampling factor. As shown, both $\epsilon$-VAE (B) and $\epsilon$-VAE (M) outperform SD-VAE across all metrics. These results confirm the effectiveness and efficiency of our one-step models compared to SD-VAE. Consequently, this allows $\epsilon$-VAE to be adapted for scenarios with latency-sensitive requirements, *e.g.*, real-time visualization during image generation, by reducing the decoding step to a single pass.

### 4.2. Class-conditional image generation

We now evaluate the generative performance of $\epsilon$-VAE when combined with latent diffusion models (Rombach et al., 2022). We perform standard class-conditional image generation tasks using the DiT-XL/2 model as our latent generative model (Peebles & Xie, 2023). Further details on the training setup are provided in Appx. C.3. Tab. 4 presents the image generation results of $\epsilon$-VAE and other competing VAEs at resolutions of $256 \times 256$. The results show that $\epsilon$-VAE consistently outperforms other VAEs across different downsampling factors. In addition, we emphasize that $\epsilon$-VAE achieves favorable generation quality while using only 25% of the token length typically required by SD-VAE.

*Table 2.* **Comparisons with state-of-the-art image autoencoders.** All results are computed on $256 \times 256$ ImageNet 50K validation set and COCO-2017 5K validation set. $\epsilon$-VAE-SD (M) achieves better reconstruction quality while having similar parameters (49M) in the decoder with other VAEs. Further improvements are obtained after we scale up to $\epsilon$-VAE-SD (H) which has 355M decoder parameters.

| Downsample factor | Method | Discrete latent | Latent dim. | ImageNet | | | COCO | | |
|---|---|---|---|---|---|---|---|---|---|
| | | | | rFID↓ | PSNR↑ | SSIM↑ | rFID↓ | PSNR↑ | SSIM↑ |
| $16 \times 16$ | VQGAN (Esser et al., 2021) | ✓ | 256 | 4.99 | 20.00 | 0.629 | 12.29 | 19.57 | 0.630 |
| | MaskGIT (Chang et al., 2022) | ✓ | 256 | 2.28 | - | - | - | - | - |
| | LlamaGen (Sun et al., 2024) | ✓ | 8 | 2.19 | 20.79 | 0.675 | 8.11 | 20.42 | 0.678 |
| | SD-VAE (Rombach et al., 2022) | ✗ | 4 | 2.93 | 20.57 | 0.662 | 8.89 | 19.95 | 0.670 |
| | $\epsilon$-VAE-SD (M) | ✗ | 4 | 1.91 | 21.27 | 0.693 | 6.12 | 22.38 | 0.718 |
| | $\epsilon$-VAE-SD (H) | ✗ | 4 | **1.35** | **22.60** | **0.711** | **4.18** | **24.26** | **0.830** |
| $8 \times 8$ | VQGAN (Esser et al., 2021) | ✓ | 4 | 1.19 | 23.38 | 0.762 | 5.89 | 23.08 | 0.771 |
| | ViT-VQGAN (Yu et al., 2022) | ✓ | 32 | 1.28 | - | - | - | - | - |
| | LlamaGen (Sun et al., 2024) | ✓ | 8 | 0.59 | 24.45 | 0.813 | 4.19 | 24.20 | 0.822 |
| | SD-VAE (Rombach et al., 2022) | ✗ | 4 | 0.74 | 25.68 | 0.820 | 4.45 | 25.41 | 0.831 |
| | SDXL-VAE (Podell et al., 2024) | ✗ | 4 | 0.68 | 26.04 | 0.834 | 4.07 | 25.76 | 0.845 |
| | LiteVAE (Sadat et al., 2024) | ✗ | 4 | 0.87 | 26.02 | 0.740 | - | - | - |
| | $\epsilon$-VAE-SD (M) | ✗ | 4 | 0.47 | 27.65 | 0.841 | 3.98 | 25.88 | 0.850 |
| | $\epsilon$-VAE-SD (H) | ✗ | 4 | **0.38** | **29.49** | **0.851** | **3.65** | **26.01** | **0.856** |

*Table 3.* **Image reconstruction results of one-step $\epsilon$-VAE and SD-VAE on ImageNet $256 \times 256$.** A downsampling factor of $8 \times 8$ is used for comparison. We include two variants of our model in the results: $\epsilon$-VAE-SD (B), which has a similar inference speed to SD-VAE, and $\epsilon$-VAE-SD (M), which matches SD-VAE in the number of parameters.

| Method | rFID↓ | PSNR↑ | SSIM↑ |
|---|---|---|---|
| SD-VAE (Rombach et al., 2022) | 0.74 | 25.68 | 0.820 |
| $\epsilon$-VAE-SD (B) | 0.57 | 25.91 | 0.826 |
| $\epsilon$-VAE-SD (M) | **0.51** | **26.45** | **0.830** |

*Table 4.* **Benchmarking class-conditional image generation on ImageNet $256 \times 256$.** We use the DiT-XL/2 architecture (Esser et al., 2024) for latent diffusion models, and we do not apply classifier-free guidance (Ho & Salimans, 2022).

| Downsample factor | Method | Throughput (image/ks) | FID↓ |
|---|---|---|---|
| $32 \times 32$ | SD-VAE (Rombach et al., 2022) | 3991 | 21.31 |
| | $\epsilon$-VAE-SD (M) | 3865 | 15.98 |
| | $\epsilon$-VAE-SD (H) | 3870 | **14.26** |
| $16 \times 16$ | SD-VAE (Rombach et al., 2022) | 1220 | 14.59 |
| | $\epsilon$-VAE-SD (M) | 1192 | 10.68 |
| | $\epsilon$-VAE-SD (H) | 1180 | **9.72** |
| $8 \times 8$ | Asym-VAE (Zhu et al., 2023) | 502 | 10.85 |
| | Omni-VAE (Wang et al., 2024b) | 480 | 12.25 |
| | SD-VAE (Rombach et al., 2022) | 522 | 11.63 |
| | $\epsilon$-VAE-SD (M) | 491 | 9.39 |
| | $\epsilon$-VAE-SD (H) | 477 | **8.85** |

This token length reduction significantly accelerates latent diffusion model generation, leading to $2.3\times$ higher inference throughput while maintaining competitive generation quality. These results confirm that the performance gains from the reconstruction task successfully transfer to the generation task, further validating the effectiveness of $\epsilon$-VAE.

More importantly, $\epsilon$-VAE-SD achieves around 25% improvement in generation quality over SD-VAE at the $32 \times 32$ downsampling factor, alongside a $3.2\times$ inference speedup than SD-VAE at the $16 \times 16$ downsampling factor with comparable FID. We observed similar training speedups for latent diffusion models utilizing $\epsilon$-VAE at this higher downsampling rate. These gains are more pronounced than those observed when increasing the downsampling factor from $8 \times 8$ to $16 \times 16$. These findings strongly suggest that the benefits of $\epsilon$-VAE and latent diffusion pipeline could be amplified with higher downsampling factors.

An additional advantage of scaling the autoencoder over the latent model lies in computational efficiency. Recent trends show latent diffusion models increasingly adopt Transformer architectures (Peebles & Xie, 2023), where self-attention scales quadratically with input resolution. In contrast, our convolution-based UNet decoder offers more favorable linear scaling. As models grow, shifting complexity to the autoencoder helps reduce the burden on the latent model, leading to a more efficient overall system.

### 4.3. Ablation studies

We conduct a component-wise analysis to validate our key design choices, focusing on three critical aspects: architecture, objectives, and noise & time scheduling. We evaluate the reconstruction quality (rFID) and sampling efficiency (NFE). The results are summarized in Tab. 5.

**Baseline.** Our evaluation begins with a baseline model: an autoencoder with a diffusion decoder, trained solely using the score matching objective. This baseline follows the vanilla diffusion setup from Ho et al. (2020), including their

*Table 5.* **Ablation study on key design choices for the $\epsilon$-VAE diffusion decoder.** A systematic evaluation of the proposed architecture ($\star$), objectives ($\dagger$), and noise & time scheduling ($\S$). Each row progressively modifies or builds upon the baseline decoder, showing improvements in performance. *The results are computed under the $\epsilon$-VAE-lite configuration.*

| Ablation | NFE↓ | rFID↓ |
|---|---|---|
| *Baseline:* DDPM-based diffusion decoder | 1,000 | 28.22 |
| $\dagger$ *(a)* Diffusion $\to$ Rectified flow parameterization | 100 | 24.11 |
| $\S$ *(b)* Uniform $\to$ Logit-normal time step sampling during training | 50 | 23.44 |
| $\star$ *(c)* DDPM UNet $\to$ ADM UNet | 50 | 22.04 |
| $\dagger$ *(d)* Perceptual matching on $\hat{x}_0^t$ and $x_0$ | 10 | 11.76 |
| $\dagger$ *(e)* Adversarial denoising trajectory matching on $(\hat{x}_0^t, x_t)$ and $(x_0, x_t)$ | 5 | 8.24 |
| $\S$ *(f)* Scale $x_t$ by $\gamma = 0.6$ | 5 | 7.08 |
| $\S$ *(g)* Uniform $\to$ Reversed logarithm time spacing during inference | 3 | 6.24 |

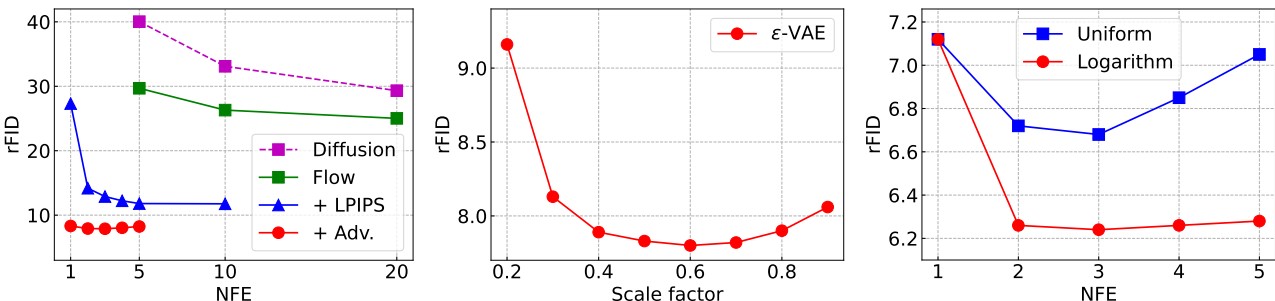

*Figure 3.* **Impact of our major diffusion decoder designs.** Improved training objectives, particularly perceptual matching loss and adversarial denoising trajectory matching loss, significantly contribute to better rFID scores and NFE (left). Effective noise scheduling by modulating the scaling factor $\gamma$ further enhances rFID, with an optimum value of 0.6 in our experiments (middle). Lastly, adjusting time step spacing during inference ensures stable sampling in low NFE regimes (right). *We report results under the $\epsilon$-VAE-lite configuration.*

UNet architecture, parameterization, and training configurations, while extending to a conditional form as described in Eq. 8. Building on this baseline, we progressively introduce updates and evaluate the impact of our proposed method.

**Impact of proposals.** In **(a)**, transitioning from standard diffusion to rectified flow (Liu et al., 2023) straightens the optimization path, resulting in significant gains in rFID and NFE. In **(b)**, adopting a logit-normal time step distribution optimizes rectified flow training (Esser et al., 2024), further improving both rFID and NFE. In **(c)**, updates to the UNet architecture (Nichol & Dhariwal, 2021) contribute to enhanced rFID scores. In **(d)**, LPIPS loss is applied to match reconstructions $\hat{x}_0^t$ with real images $x_0$. In **(e)**, adversarial trajectory matching loss aligns $(\hat{x}_0^t, x_t)$ with $(x_0, x_t)$, the target transition in rectified flow. Both objectives improve model understanding of the underlying optimization trajectory, significantly enhancing rFID scores and NFE.

Up to this point, with the full implementation of Eq. 1, we can compare our proposal with the VAE (B) model, which achieves an rFID score of 11.15. Our model, with a score of 8.24, already surpasses this baseline. We further improve performance by optimizing noise and time scheduling within our framework, as described next.

In **(f)**, scaling $x_t$ reduces the signal-to-noise ratio (Chen, 2023), presenting challenges for more effective learning during training. Fig. 3 (middle) demonstrates that a scaling factor of 0.6 produces the best results. Finally, in **(g)**, reversed logarithmic time step spacing during inference allows for denser evaluations in noisier regions. Fig. 3 (right) demonstrates that this method provides more stable sampling in the lower NFE regime compared to the original uniform spacing.

In Fig. 3 (right), reconstruction quality degrades when the number of denoising steps exceeds three. To enable large step sizes for the reverse process during inference, we introduce the denoising trajectory matching loss to implicitly model the conditional distribution $p(x_0|x_t)$, shifting the denoising distributions from traditional Gaussian to non-Gaussian multimodal forms (Xiao et al., 2022). However, the assumptions underlying this approach are most effective when the total number of denoising steps is small. This reveals an optimal range of one to three inference steps. The degradation beyond this range also suggests that uniform step spacing may no longer be ideal. Accordingly, we empirically explored alternative sampling strategies and found that a reversed logarithmic schedule yields improved performance, as shown in the figure.

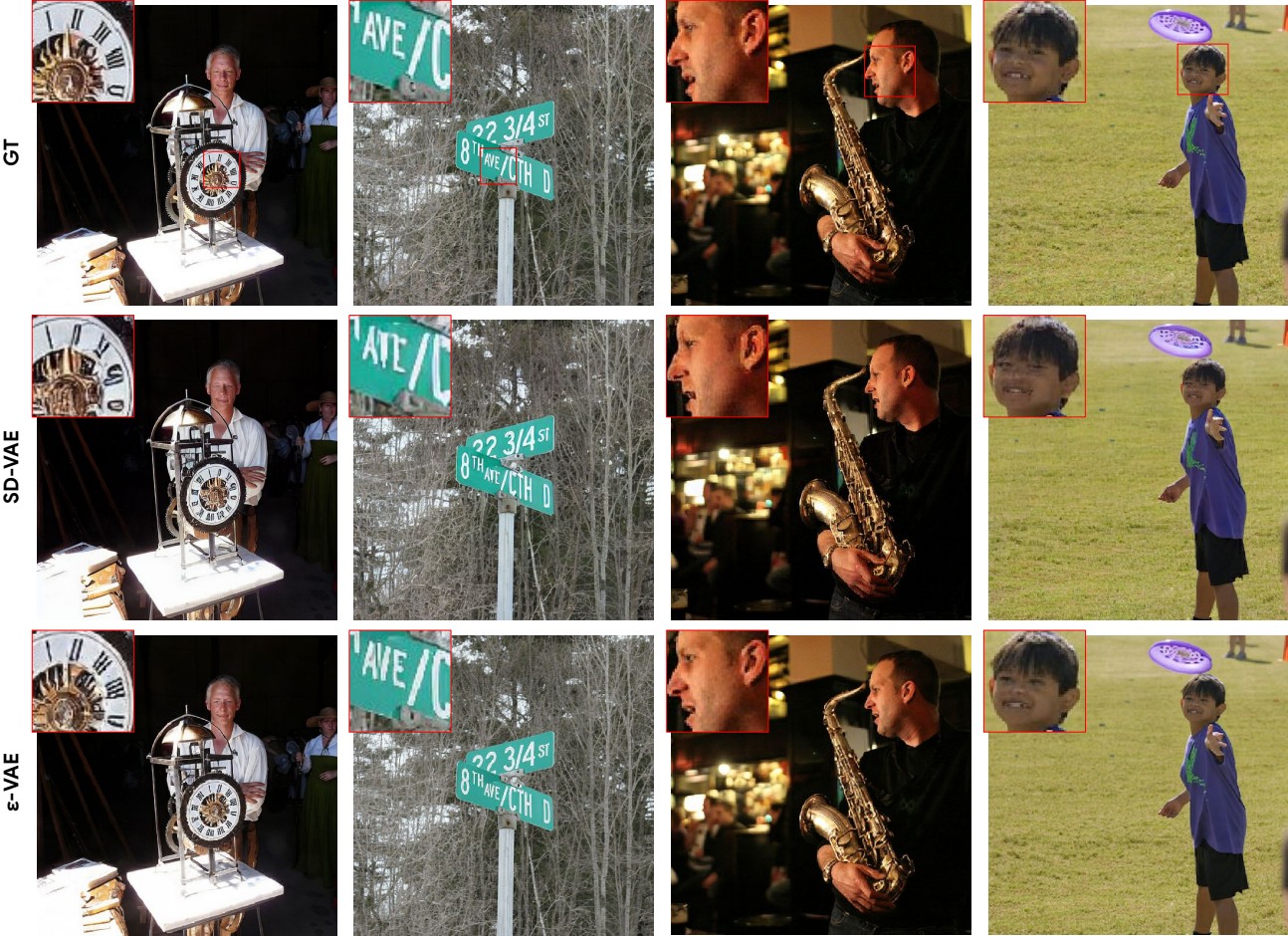

*Figure 4.* **Image reconstruction results under the SD-VAE configuration** ([Rombach et al., 2022](#)) **at the resolution of** $512 \times 512$. We find that $\epsilon$-VAE produces more accurate visual details than SD-VAE in the highlighted regions with text or human face. *Best viewed when zoomed-in and in color.*

### 4.4. Visualization

In addition to the quantitative results, Fig. 4 shows high-resolution image reconstruction samples produced by SD-VAE ([Rombach et al.,](#) 2022) and $\epsilon$-VAE at the resolution of $512 \times 512$. We observe that reconstructed images generated by $\epsilon$-VAE demonstrate a better visual quality than ones of SD-VAE. In particular, $\epsilon$-VAE maintains a good visual quality for small text and human face. We provide more visual comparisons in Appx. E and throughout discussions on the major properties and advantages of $\epsilon$-VAE compared to traditional VAEs in Appx. A.

## 5. Conclusion

We present $\epsilon$-VAE, an effective visual tokenizer that introduces a diffusion decoder into standard autoencoders, turning single-step decoding into a multi-step probabilistic process. By exploring key design choices in modeling, objectives, and diffusion training, we demonstrate significant performance improvements. Our approach outperforms traditional autoencoders in both reconstruction and generation quality, particularly in high-compression scenarios. We hope our concept of iterative generation during decoding inspires further advancements in visual autoencoding.

## Acknowledgements

We would like to thank Xingyi Zhou, Weijun Wang, and Caroline Pantofaru for reviewing the paper and providing feedback. We thank Rui Qian, Xuan Yang, and Mingda Zhang for helpful discussion. We also thank the Google Kauldron team for technical assistance.

## Impact statement

Our work could lead to improved autoencoding techniques which have the potential to benefit generative modeling

across various perspectives, including reducing training time and memory requirements, improving visual qualities, *etc.* Although our work does not uniquely raise any new ethical challenges, visual generative modeling is a field with several ethical concerns worth acknowledging. For example, there are known issues around bias and fairness, either in the representation of generated images (Menon et al., 2020) or the implicit encoding of stereotypes (Steed & Caliskan, 2021), as well as potential risks in privacy. To ensure that the benefits of this technology are harnessed responsibly, we encourage continued open discussions in the community around the development of these new technologies.

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

# A. Discussion

**Distribution-aware compression.** Traditional image compression methods optimize the rate-distortion trade-off (Shannon et al., 1959), prioritizing compactness over input fidelity. Building on this, we also aim to capture the broader input distribution during compression, generating compact representations suitable for latent generative models. This approach introduces an additional dimension to the trade-off, perception or distribution fidelity (Blau & Michaeli, 2018), which aligns more closely with the rate-distortion-perception framework (Blau & Michaeli, 2019).

**Iterative and stochastic decoding.** A key question within the rate-distortion-perception trade-off is whether the iterative, stochastic nature of diffusion decoding offers advantages over traditional single-step, deterministic methods (Kingma, 2013). The strengths of diffusion (Ho et al., 2020) lie in its iterative process, which progressively refines the latent space for more accurate reconstructions, while stochasticity allows for capturing complex variations within the distribution. Although iterative methods may appear less efficient, our formulation is optimized to achieve optimal results in just three steps and also supports single-step decoding, ensuring decoding efficiency remains practical (see Fig. 3 (left)). While stochasticity might suggest the risk of "hallucination" in reconstructions, the outputs remain faithful to the underlying distribution by design, producing perceptually plausible results. This advantage is particularly evident under extreme compression scenarios (see Fig. 5), with the degree of stochasticity adapting based on compression levels (see Fig. 6).

**Multi-step *vs.* single-step decoding.** While replacing single-step decoding with an iterative process may seem counter-intuitive due to increased computational cost, the diffusion-based decoder addresses this concern in three key ways. First, it offers scalable inference, where even a single-step variant already outperforms a plain VAE decoder, and additional steps further enhance quality (see Tab. 3). Second, it provides controllable trade-offs between computation and visual fidelity, allowing the number of steps to be adjusted at inference time based on application needs. Third, as shown in Tab. 4, it enables training under higher compression ratios, which helps offset the added cost of iterative decoding by reducing the size of latent representations.

**Scalability.** As discussed in Sec. 4.1, our diffusion-based decoding method maintains the resolution generalizability typically found in standard autoencoders. This feature is highly practical: the autoencoder is trained on lower-resolution images, while the subsequent latent generative model is trained on latents derived from higher-resolution inputs. However, we acknowledge that memory overhead and throughput become concerns with our UNet-based diffusion decoder, especially for high-resolution inputs. This challenge becomes more pronounced as models, datasets, or resolutions scale up. A promising future direction is patch-based diffusion (Ding et al., 2024; Wang et al., 2024c), which partitions the input into smaller, independently processed patches. This approach has the potential to reduce memory usage and enable faster parallel decoding.

# B. Related work

**Image tokenization.** Image tokenization is crucial for effective generative modeling, transforming images into compact, structured representations. A common approach employs an autoencoder framework (Hinton & Salakhutdinov, 2006), where the encoder compresses images into low-dimensional latent representations, and the decoder reconstructs the original input. These latent representations can be either discrete commonly used in autoregressive models (Van den Oord et al., 2016; Van Den Oord et al., 2017; Chen et al., 2020; Chang et al., 2022; Yu et al., 2023; Kondratyuk et al., 2024), or continuous, as found in diffusion models (Ho et al., 2020; Dhariwal & Nichol, 2021; Rombach et al., 2022; Peebles & Xie, 2023; Gupta et al., 2023; Brooks et al., 2024). The foundational form of visual autoencoding today originates from Van Den Oord et al. (2017). While advancements have been made in modeling (Yu et al., 2022; 2024b; Chen et al., 2024), objectives (Zhang et al., 2018; Karras et al., 2019; Esser et al., 2021), and quantization methods (Yu et al., 2024a; Zhao et al., 2024), the core encoding-and-decoding scheme remains largely the same.

In this work, we propose a new perspective by replacing the traditional decoder with a diffusion process. Specifically, our new formulation retains the encoder but introduces a conditional diffusion decoder. Within this framework, we systematically study various design choices, resulting in a significantly enhanced autoencoding setup.

Additionally, we refer to the recent work MAR (Li et al., 2024), which leverages diffusion to model per-token distribution in autoregressive frameworks. In contrast, our approach models the overall input distribution in autoencoders using diffusion. This difference leads to distinct applications of diffusion during generation. For instance, MAR generates samples autoregressively, decoding each token iteratively using diffusion, token by token. In our method, we first sample all tokens from the downstream generative model and then decode them iteratively using diffusion as a whole.

**Image compression.** Our work shares similarities with recent image compression approaches that leverage diffusion models. For example, Hoogeboom et al. (2023a); Birodkar et al. (2024) use diffusion to refine autoencoder residuals, enhancing high-frequency details. Yang & Mandt (2024) employs a diffusion decoder conditioned on quantized dis-

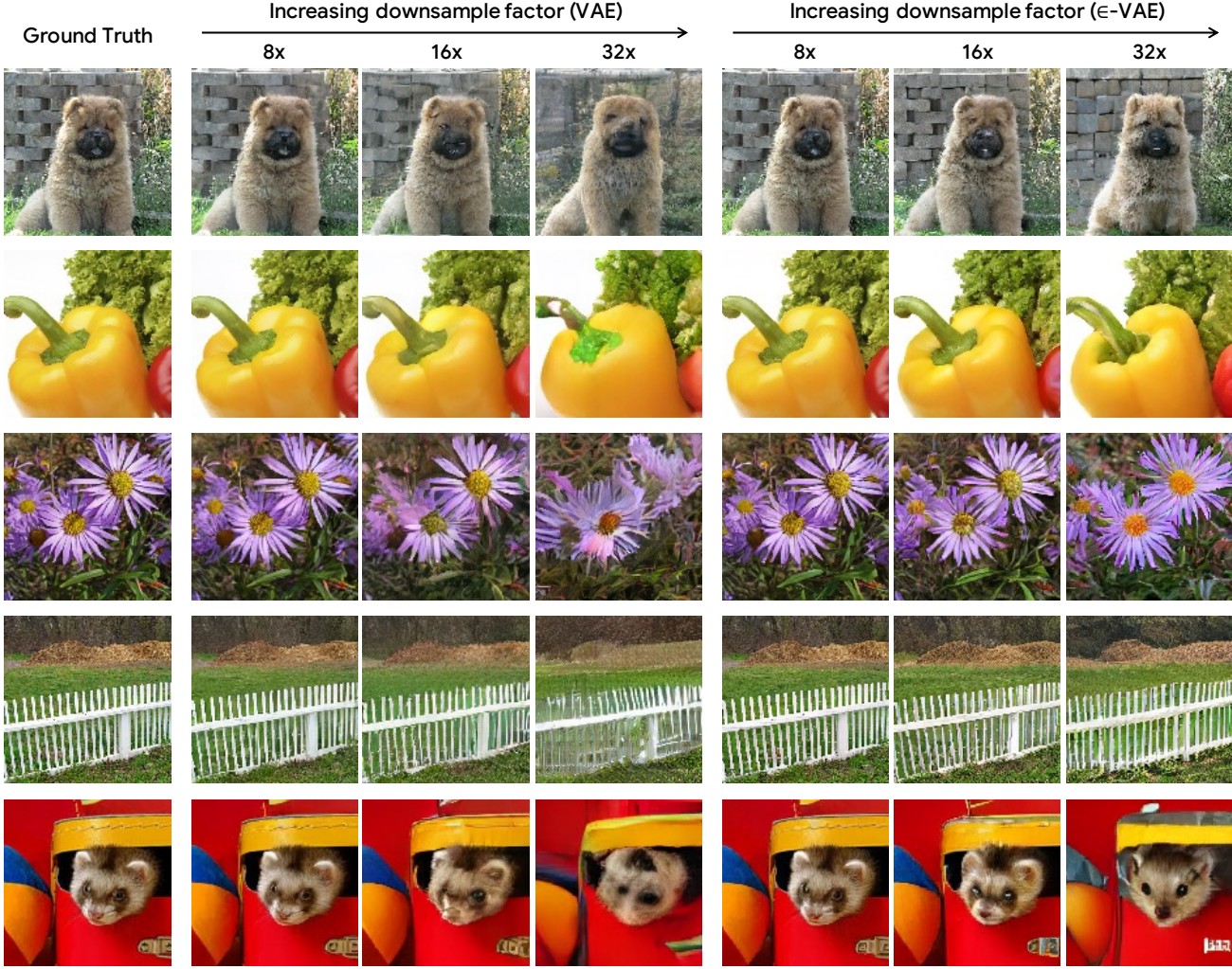

*Figure 5.* **Reconstruction results with varying downsampling ratios.** $\epsilon$-VAE maintains both high fidelity and perceptual quality, even under extreme downsampling conditions, whereas VAE fails to preserve semantic integrity. *Best viewed when zoomed-in and in color.*

crete codes and omits the GAN loss. However, these methods primarily focus on the traditional rate-distortion tradeoff, balancing rate (compactness) and distortion (input fidelity) (Shannon et al., 1959), with the goal of storing and transmitting data efficiently without significant loss of information.

In this work, we emphasize perception (distribution fidelity) alongside the rate-distortion tradeoff, ensuring that reconstructions more closely align with the overall data distribution (Heusel et al., 2017; Zhang et al., 2018; Blau & Michaeli, 2019), thereby enhancing the decoded results from the sampled latents of downstream generative models. We achieve this by directly integrating the diffusion process into the decoder, unlike Hoogeboom et al. (2023a); Birodkar et al. (2024). Moreover, unlike Yang & Mandt (2024), we do not impose strict rate-distortion regularization in the latent space and allow the GAN loss to synergize with our approach.

**Diffusion decoder.** Several studies (Preechakul et al., 2022; Shi et al., 2022; Pernias et al., 2024; Nguyen & Tran, 2024; Sauer et al., 2024; Luo et al., 2023) have explored diffusion decoders conditioned on compressed latents of the input, which are relevant to our work. We outline the key differences between these works and $\epsilon$-VAE: First, prior works have not fully leveraged the synergy between diffusion decoders and standard VAE training objectives. In this work, we enhance state-of-the-art VAE objectives by replacing the reconstruction loss with a score matching loss and adapting LPIPS and GAN losses to ensure compatibility with diffusion decoders. These changes yield significant improvements in autoencoding performance, as evidenced by lower rFID scores and faster inference. Second, we are the first to investigate various parameterizations (*e.g.*, epsilon and velocity) and demonstrate that modern velocity parameterization, coupled with optimized train and test-time

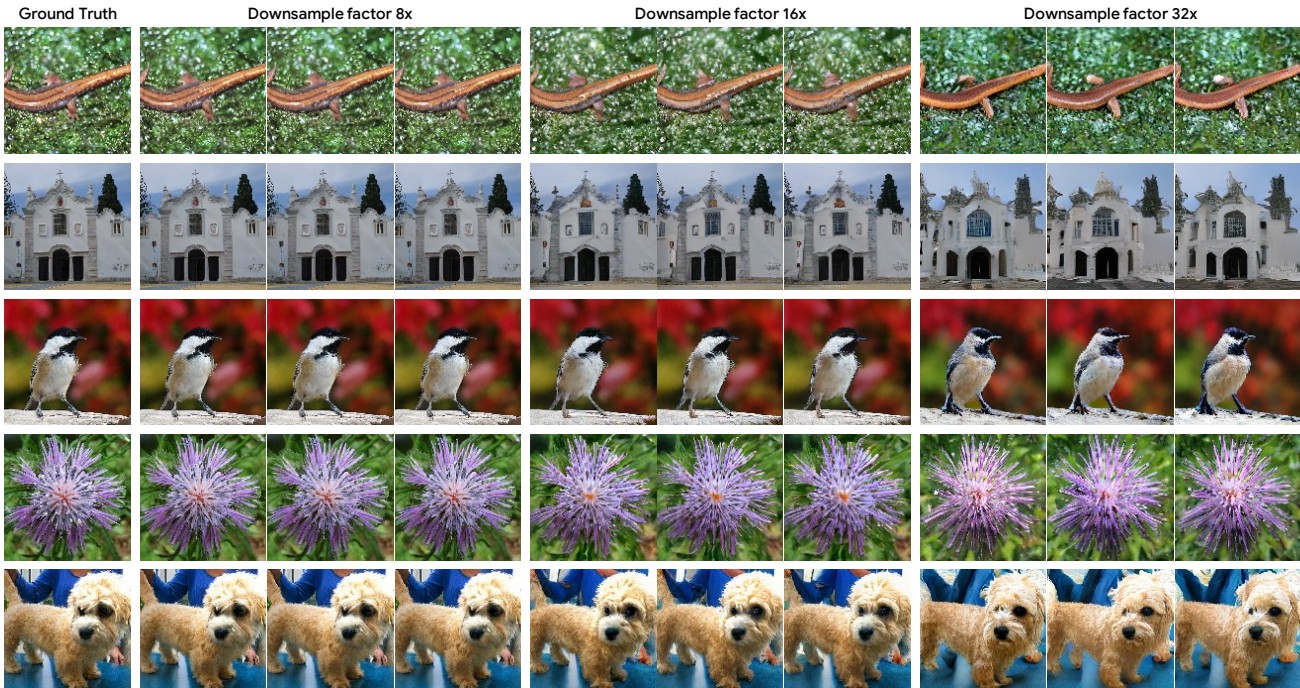

*Figure 6.* *ε*-**VAE reconstruction results with varying random seeds and downsampling ratios.** At lower compression levels, the reconstruction behaves more deterministically, whereas higher compression introduces stochasticity, enabling more flexible reconstruction of plausible inputs. *Best viewed when zoomed-in and in color.*

noise scheduling, provides substantial benefits. These enhancements improve both reconstruction performance and sampling efficiency. Third, previous diffusion-based decoders (Preechakul et al., 2022; Shi et al., 2022; Pernias et al., 2024), which often rely on ad-hoc techniques like distillation or consistency regularization to speed up inference (Nguyen & Tran, 2024; Sauer et al., 2024; Luo et al., 2023), our approach achieves fast decoding (1 to 3 steps) without such techniques. This is made possible by integrating our proposed objectives and parameterizations. Last but not least, $\epsilon$-VAE exhibits strong resolution generalization capabilities, a key property of standard VAEs. In contrast, models like DiffusionAE (Preechakul et al., 2022) and DiVAE (Shi et al., 2022) either lack this ability or are inherently limited. For example, DiVAE's bottleneck add/concat design restricts its capacity to generalize across resolutions.

SWYCC (Birodkar et al., 2024) also explores joint learning of continuous encoders and decoders using a diffusion model. However, SWYCC differs fundamentally from our approach: it replaces the GAN loss with a diffusion-based loss, while we focus on identifying optimal synergies between traditional autoencoding losses (including GAN loss) and diffusion-based decoding. Our goal is to identify an optimal strategy for combining these elements, rather than simply substituting one for another.

Another closely related work, DiTo (Chen et al., 2025), also presents a diffusion-based tokenizer which learns compact visual representations for image generation. Its main insight is that a single diffusion learning objective is capable of training scalable image tokenizers. More than that, our method demonstrates that traditional autoencoding losses such as LPIPS and GAN losses are complimentary to the diffusion target, leading to better reconstruction quality. This design substantially differ our work from DiTo.

While following a different motivation, Lee et al. (2023) essentially also proposes a VAE with a denoising decoder but uses the encoding as the "initial noise" instead of as conditioning for a standard diffusion model starting from a standard Gaussian distribution. This idea could be potentially used for speeding up the proposed approach, which we will explore in the future.

**Image generation.** Recent advances in image generation span a wide range of approaches, including VAEs (Kingma, 2013), GANs (Goodfellow et al., 2014), autoregressive models (Chen et al., 2020) and diffusion models (Song et al., 2021; Ho et al., 2020). Among these, diffusion models have emerged as the leading approach for generating high-dimensional data such as images (Saharia et al., 2022a; Baldridge et al., 2024; Esser et al., 2024) and videos (Brooks et al., 2024; Gupta et al., 2023), where the gradual refinement of global structure is crucial. The current focus in diffusion-based generative models lies in advancing archi-

*Table 6.* **Hyper-parameters for decoder variants.**

| Model | Channel dim. | Depth multipliers | # of blocks |
|---|---|---|---|
| Base (B) | 64 | $\{1, 1, 2, 2, 4\}$ | 2 |
| Medium (M) | 96 | $\{1, 1, 2, 2, 4\}$ | 2 |
| Large (L) | 128 | $\{1, 1, 2, 2, 4\}$ | 2 |
| Extra-large (XL) | 128 | $\{1, 1, 2, 2, 4\}$ | 4 |
| Huge (H) | 256 | $\{1, 1, 2, 2, 4\}$ | 2 |

*Table 7.* **Image reconstruction results on ImageNet** $128 \times 128$.

| Configuration | NFE↓ | rFID↓ |
|---|---|---|
| *Baseline (c)* in Tab. 5: | | |
|   Inject conditioning by channel-wise concatenation | 50 | 22.04 |
|   Inject conditioning by AdaGN | 50 | 22.01 |
| *Baseline (e)* in Tab. 5: | | |
|   Matching the distribution of $\hat{x}_0^t$ and $x_0$ | - | N/A |
|   Matching the trajectory of $x_t \rightarrow x_0$ | 5 | 8.24 |
|   Matching the trajectory of $x_t \rightarrow x_{t-\Delta t}$ | 5 | 10.53 |

tectures (Rombach et al., 2022; Peebles & Xie, 2023; Hoogeboom et al., 2023b), parameterizations (Karras et al., 2022; Kingma & Gao, 2024; Ma et al., 2024; Esser et al., 2024), or better training dynamics (Nichol & Dhariwal, 2021; Chen, 2023; Chen et al., 2023). However, tokenization, an essential component in modern diffusion models, often receives less attention.

In this work, we focus on providing compact continuous latents without applying quantization during autoencoder training (Rombach et al., 2022), as they have been shown to be effective in state-of-the-art latent diffusion models (Rombach et al., 2022; Saharia et al., 2022a; Peebles & Xie, 2023; Esser et al., 2024; Baldridge et al., 2024). We compare our autoencoding performance against the baseline approach (Esser et al., 2021) using the DiT framework (Peebles & Xie, 2023) as the downstream generative model.

## C. Experiment setups

In this section, we provide additional details on our experiment configurations for reproducibility.

### C.1. Model specifications

Tab. 6 summarizes the primary architecture details for each decoder variant. The channel dimension is the number of channels of the first U-Net layer, while the depth multipliers are the multipliers for subsequent resolutions. The number of residual blocks denotes the number of residual stacks contained in each resolution.

### C.2. Implementation details

During the training of discriminators, Esser et al. (2021) introduced an adaptive weighting strategy for $\lambda_{adv}$. However, we notice that this adaptive weighting does not introduce any benefit which is consistent with the observation made by Sadat et al. (2024). Thus, we set $\lambda_{adv} = 0.5$ in the experiments for more stable model training across different configurations.

The autoencoder loss follows Eq. 1, with weights set to $\lambda_{LPIPS} = 0.5$ and $\lambda_{adv} = 0.5$. We use the Adam optimizer (Kingma & Ba, 2015) with $\beta_1 = 0$ and $\beta_2 = 0.999$, applying a linear learning rate warmup over the first 5,000

steps, followed by a constant rate of 0.0001 for a total of one million steps. The batch size is 256, with data augmentations including random cropping and horizontal flipping. An exponential moving average of model weights is maintained with a decay rate of 0.999. All models are implemented in JAX/Flax (Bradbury et al., 2018; Heek et al., 2024) and trained on TPU-v5lite pods.

### C.3. Latent diffusion models

We follow the setting in Peebles & Xie (2023) to train the latent diffusion models for unconditional image generation on the ImageNet dataset. The DiT-XL/2 architecture is used for all experiments. The diffusion hyperparameters from ADM (Dhariwal & Nichol, 2021) are kept. To be specific, we use a $t_{max} = 1000$ linear variance schedule ranging from 0.0001 to 0.02, and results are generated using 250 DDPM sampling steps. For simplicity and training stability, we remove the variational lower bound loss term during training, which leads to a slight drop in generation qualities.

All models are trained with Adam (Kingma & Ba, 2015) with no weight decay. We use a constant learning rate of 0.0001 and a batch size of 256. Horizontal flipping and random cropping are used for data augmentation. We maintain an exponential moving average of DiT weights over training with a decay of 0.9999. We use identical training hyperparameters across all experiments and train models for one million steps in total. No classifier-free guidance (Ho & Salimans, 2022) is employed in all the experiments. Inference throughputs are computed on a Tesla H100 GPU.

## D. Additional experimental results

We note that all experiments conducted in this section are under the $\epsilon$-VAE-lite configuration.

**Conditioning.** In addition to injecting conditioning via channel-wise concatenation, we explore providing conditioning to the diffusion model by adaptive group normalization (AdaGN) (Nichol & Dhariwal, 2021; Dhariwal & Nichol, 2021). To achieve this, we resize the conditioning (*i.e.*, encoded latents) via bilinear sampling to the desired

*Table 8.* **Model scaling and resolution generalization analysis.** Five model variants are trained and evaluated. $\Delta_{\text{rFID}}$ represents the absolute differences (or relative ratio) in rFID between the corresponding model size variants of VAE and $\epsilon$-VAE. [†] denotes resolution generalization experiments. To fairly evaluate the impact of $\epsilon$-VAE under controlled model parameters, we highlight three groups of model variants with comparable parameters, using different colors.

| Model | $\mathcal{G}$ params (M) | **ImageNet** $128 \times 128$ | | **ImageNet** $256 \times 256$ [†] | | **ImageNet** $512 \times 512$ [†] | |
|---|---|---|---|---|---|---|---|
| | | **rFID**↓ | $\Delta_{\text{rFID}}$ | **rFID**↓ | $\Delta_{\text{rFID}}$ | **rFID**↓ | $\Delta_{\text{rFID}}$ |
| VAE (B) | 10.14 | 11.15 | - | 5.74 | - | 3.69 | - |
| VAE (M) | 22.79 | 9.26 | - | 4.63 | - | 2.69 | - |
| VAE (L) | 40.48 | 8.49 | - | 4.78 | - | 2.78 | - |
| VAE (XL) | 65.27 | 7.58 | - | 4.42 | - | 2.41 | - |
| VAE (H) | 161.81 | 7.12 | - | 4.29 | - | 2.37 | - |
| $\epsilon$-VAE (B) | 20.63 | 6.24 | 4.91 (44.0%) | 3.90 | 1.84 (32.0%) | 2.06 | 1.63 (44.2%) |
| $\epsilon$-VAE (M) | 49.33 | 5.42 | 3.84 (41.5%) | 2.79 | 1.84 (39.7%) | 2.02 | 0.67 (24.9%) |
| $\epsilon$-VAE (L) | 88.98 | 4.71 | 3.78 (44.5%) | 2.60 | 2.03 (43.8%) | 1.92 | 0.86 (30.9%) |
| $\epsilon$-VAE (XL) | 140.63 | 4.18 | 3.40 (44.9%) | 2.38 | 2.04 (46.2%) | 1.82 | 0.59 (24.5%) |
| $\epsilon$-VAE (H) | 355.62 | 4.04 | 3.08 (43.3%) | 2.31 | 1.98 (46.2%) | 1.78 | 0.59 (24.9%) |

*Table 9.* **Unconditional image generation quality.** The DiT-XL/2 is trained on latents provided by the trained autoencoders, VAE and $\epsilon$-VAE, with varying model sizes using ImageNet. We evaluate the generation quality at resolutions of $128 \times 128$ and $256 \times 256$ using four standard metrics. Additionally, we report rFID to determine if the improvement trend observed in reconstruction task extends to the generation task. We highlight three groups of model variants with comparable parameters.

| Model | **ImageNet** $128 \times 128$ | | | | | **ImageNet** $256 \times 256$ | | | | |
|---|---|---|---|---|---|---|---|---|---|---|
| | **rFID**↓ | **FID**↓ | **IS**↑ | **Prec.**↑ | **Rec.**↑ | **rFID**↓ | **FID**↓ | **IS**↑ | **Prec.**↑ | **Rec.**↑ |
| VAE (B) | 11.15 | 36.8 | 17.9 | 0.48 | 0.53 | 5.74 | 46.6 | 23.4 | 0.45 | 0.56 |
| VAE (M) | 9.26 | 34.6 | 18.2 | 0.49 | 0.55 | 4.63 | 44.7 | 23.8 | 0.47 | 0.58 |
| VAE (L) | 8.49 | 33.9 | 18.4 | 0.50 | 0.56 | 4.78 | 44.3 | 24.7 | 0.47 | 0.59 |
| VAE (XL) | 7.58 | 31.7 | 19.3 | 0.51 | 0.57 | 4.42 | 43.1 | 24.9 | 0.47 | 0.59 |
| VAE (H) | 7.12 | 30.9 | 19.8 | 0.52 | 0.57 | 4.29 | 41.6 | 25.9 | 0.48 | 0.59 |
| $\epsilon$-VAE (B) | 6.24 | 29.5 | 20.7 | 0.53 | 0.59 | 3.90 | 39.5 | 25.2 | 0.46 | 0.61 |
| $\epsilon$-VAE (M) | 5.42 | 27.6 | 21.2 | 0.55 | 0.59 | 2.79 | 35.4 | 26.2 | 0.51 | 0.62 |
| $\epsilon$-VAE (L) | 4.71 | 27.3 | 22.1 | 0.55 | 0.59 | 2.60 | 34.8 | 26.5 | 0.51 | 0.63 |
| $\epsilon$-VAE (XL) | 4.18 | 25.3 | 22.7 | 0.55 | 0.59 | 2.38 | 34.0 | 27.4 | 0.53 | 0.63 |
| $\epsilon$-VAE (H) | 4.04 | 24.9 | 23.0 | 0.56 | 0.60 | 2.31 | 33.2 | 27.5 | 0.54 | 0.64 |

resolution of each stage in the U-Net model, and incorporates it into each residual block after a group normalization operation (Wu & He, 2018). This is similar to adaptive instance norm (Karras et al., 2019) and FiLM (Perez et al., 2018). We report the results in Tab. 7 (top), where we find that channel-wise concatenation and AdaGN obtain similar reconstruction quality in terms of rFID. Because of the additional computational cost required by AdaGN, we thus apply channel-wise concatenation in our model by default.

**Trajectory matching.** The proposed denoising trajectory matching objective matches the start-to-end trajectory $x_t \to x_0$ by default. One alternative choice is to directly matching the distribution of $\hat{x}_0^t$ and $x_0$ without coupling on $x_t$. However, we find this formulation leads to unstable training and could not produce reasonable results. Here, we present the results when matching the trajectory of $x_t \to x_{t-\Delta t}$, which is commonly used in previous work (Xiao et al., 2022; Wang et al., 2024a). Specifically, for each timestep $t$ during training, we randomly sample a

step $\Delta t$ from $(0, t)$. Then, we construct the real trajectory by computing $x_{t-\Delta t}$ via Eq. 5 and concatenating it with $x_t$, while the fake trajectory is obtained in a similar way but using Eq. 9 instead. Tab. 7 (bottom) shows the comparison. We observe that matching trajectory $x_t \to x_0$ yields better performance than matching trajectory $x_t \to x_{t-\Delta t}$, confirming the effectiveness of the proposed objective which is designed for the rectified flow formulation.

**Comparisons with plain diffusion ADM.** Under the same training setup of Tab. 5, we directly trained a plain diffusion model (ADM) for comparison, which resulted in rFID score of 38.26. Its conditional form is already provided as a baseline in Tab. 5, achieving 28.22. This demonstrates that our conditional form $p(x_{t-1}|x_t, z)$ offers a better approximation of the true posterior $q(x_{t-1}|x_t, x_0)$ compared to the standard form $p(x_{t-1}|x_t)$. By further combining LPIPS and GAN loss, we achieve rFID of 8.24, outperforming its VAE counterpart, which achieves 11.15. With better training configurations, our final rFID improves to 6.24. This pro-

| GT | SD-VAE | ε-VAE | GT | SD-VAE | ε-VAE |
| --- | --- | --- | --- | --- | --- |

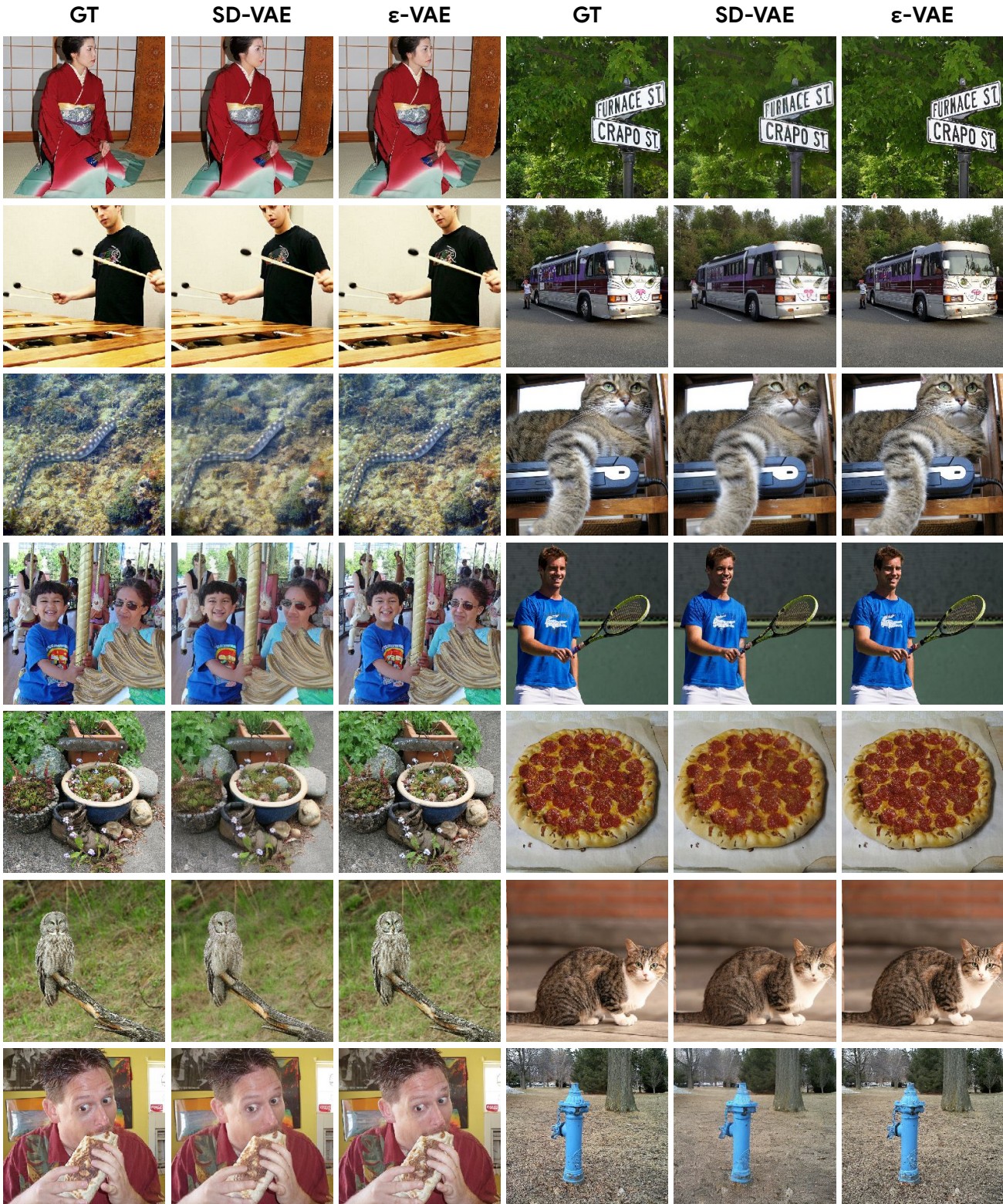

*Figure 7.* **Image reconstruction results under the SD-VAE configuration (Rombach et al., 2022) at the resolution of** $256 \times 256$**.**
$\epsilon$-VAE produces significantly better visual details than SD-VAE when reconstructing local regions with complex textures or structures, such as human faces and small texts. *Best viewed when zoomed-in and in color.*

*Table 10.* **Benchmarking class-conditional image generation on ImageNet** $256 \times 256$. We use the DiT-XL/2 architecture (Esser et al., 2024) for latent diffusion models and apply classifier-free guidance (Ho & Salimans, 2022).

| Downsample factor | Method | FID↓ |
|---|---|---|
| | SD-VAE (Rombach et al., 2022) | 3.51 |
| $8 \times 8$ | $\epsilon$-VAE-SD (M) | 2.83 |
| | $\epsilon$-VAE-SD (H) | **2.69** |

gression, from plain diffusion ADM to $\epsilon$-VAE, underscores the significance of our proposals and their impact.

**Model scaling.** We investigate the impact of model scaling by comparing VAE and $\epsilon$-VAE across five model variants, all trained and evaluated at a resolution of $128 \times 128$, as summarized in Tab. 8. The results demonstrate that $\epsilon$-VAE consistently achieves significantly better rFID scores than VAE, with an average relative improvement of over $40\%$, and even the smallest $\epsilon$-VAE model outperforms VAE at largest scale. While the U-Net-based decoder of $\epsilon$-VAE has about twice as many parameters as standard decoder of VAE, grouping models by similar sizes, highlighted in different colors, shows that performance gains are not simply due to increased model parameters.

Tab. 9 presents the unconditional image generation results of VAE and $\epsilon$-VAE at resolutions of $128 \times 128$ and $256 \times 256$. In addition to FID, we report Inception Score (IS) (Salimans et al., 2016) and Precision/Recall (Kynkäänniemi et al., 2019) as secondary metrics. The results show that $\epsilon$-VAE consistently outperforms VAE across all model scales. Notably, $\epsilon$-VAE (B) surpasses VAE (H), consistent with our earlier findings in Sec. 4.1. These results further demonstrate the effectiveness of $\epsilon$-VAE from the generation perspective.

**Results with classifier-free guidance.** We provide additional results with classifier-free guidance (Ho & Salimans, 2022) under the $8 \times 8$ downsample factor in Tab. 10. We find that $\epsilon$-VAE (M) performs relatively 20% better than SD-VAE and further improvements are obtained after we scale up our model to $\epsilon$-VAE (H). These results are consistent with the results without classifier-free guidance in Tab. 4, confirming the effectiveness of our model.

# E. Additional visual results

We provide additional visual comparisons between $\epsilon$-VAE and SD-VAE at the resolution of $256 \times 256$ (Fig. 7). Our observations indicate that $\epsilon$-VAE delivers significantly better visual quality than SD-VAE, particularly when reconstructing local regions with complex textures or structures, such as human faces and small text.

