# OpenReview forum: "Epsilon-VAE: Denoising as Visual Decoding"
_ICML.cc/2025/Conference — ICML 2025 poster_

### Official Review · Reviewer_3ACU · 2025-03-07

**Overall Recommendation:** 4

**Summary:**

- The paper proposes an alternative autoencoder that itself uses a (conditional) diffusion / rectified flow model as decoder replacing the standard VAE for latent diffusion models (LDMs).
- For that, the paper experimentally explores the design space in terms of decoder / denoiser architecture, the implementation of the conditioning, the training objective as a combination of different loss functions, the sampling of noise levels during training, and the time schedule for sampling at test time. Within this exploration, the authors propose
  - a specific combination of loss terms including a novel adversarial trajectory matching loss replacing the patch-wise GAN loss in the usual VAE training for LDMs,
  - a reversed logarithmic mapping for the timesteps during sampling at test time to have denser steps early in the inference process.
- The experimental evaluation shows advantages in reconstruction over other VAE baselines, which also translate to advantages in generation FID when combinated with a LDM. The approach enables a 4x stronger spatial compression with comparable FID than the well established StableDiffusion VAE + LDM pipeline, which results in a 2.3x increase of throughput (image/sec).

**Claims And Evidence:**

Most claims are supported by convincing evidence except for:
- The authors claim that the approach "enhances downstream generation quality by 22% and provides 2.3x inference speedup" (lines 32 ff., left column), which can be misunderstood as the experimental evaluation only shows that either the first or the second can be achieved but not both at the same time, by changing the downsampling factor of the autoencoder. It should be made clear that it achieves 22% better generation quality for the same downsampling factor or 2.3x inference speedup with comparable FID by increasing the downsampling factor.
- The authors "anticipate two key improvements" with one being "more effective generation of latent representations, allowing the downstream latent diffusion model to learn more efficiently" (lines 142 ff., left column). The paper only provides evidence for improved generation quality, but it remains unclear whether this is because of a higher level of detail achieved by the denoising-based VAE decoder or whether the different latent space also enables more efficient training of the LDM, e.g., in terms of faster convergence.

**Essential References Not Discussed:**

I am not aware of any essential missing references. Many related works are discussed in the appendix, but not in the main paper. It might make sense to briefly describe the most relevant ones in the main paper, e.g., [1].
A later version of the paper could discuss the concurrent work mentioned in the above review section.

[1] Würstchen: An efficient architecture for large-scale text-to-image diffusion models. ICLR 2024

**Experimental Designs Or Analyses:**

All experimental designs and analyses seem to be valid.

**Methods And Evaluation Criteria:**

Both methods and evaluation criteria make sense for the problem at hand:
- The paper extensively explores and ablates the design choices for their approach with all of them contributing to the final performance.
- The evaluation uses ImageNet in different resolutions and COCO as well as (r)FID, PSNR, and SSIM, which are established benchmark datasets and metrics for image reconstruction and generation.

**Other Comments Or Suggestions:**

No other comments or suggestions

**Other Strengths And Weaknesses:**

Strengths:
- The paper is well-written and easy to understand.
- The experimental results are convincing.
  - The proposed approach achieves improvements in reconstruction, which are shown to translate to improvements in generation when combined with a LDM.
  - It enables a larger downsample factor than the standard VAE while providing comparable (even slightly better) generation quality.
  - The extensive ablation study (table 4) validates the effectiveness of all design choices.
- The paper introduces additional technical contributions like the adversarial denoising trajectory matching and the reversed logarithm time spacing during inference.

Weaknesses:
- Because of the denoising procedure with 3 network forward evaluations used in the paper, the decoding is more expensive than with a standard VAE, which the paper misses to openly discuss. This limits its application in cases with real-time visualization, e.g., during the generation process of images.
- The approach seems to be limited in terms of downsampling factors and leveraging the full potential of the denoising generative paradigm:
  - The main paper misses to explore even larger compression factors in combination with training a LDM, which could be interesting to see whether even more efficient training of the expensive diffusion model is possible.
  - For both uniform or logarithmic spacing of timesteps for sampling, the reconstruction FID degrades for more than 3 denoising steps which is counterintuitive (see figure 3).
- The paper is partially a bit repetitive:
  - Velocity prediction (lines 145 ff., right column) and eq. (9) already discussed before together with rectified flows (lines 102 ff., right column) and eq. (6).
  - Noise scheduling (lines 199 ff., right column) is related work that also has been partially addressed already in the related work section.
  - Model configurations (lines 238 ff., right column) and decoder architecture (lines 312 ff., left column) overlap in content.
- The qualitative comparisons in the paper and the appendix are not very convincing showing only very minor differences in high-frequency details.
- The proposed reversed logarithm mapping for the timesteps during sampling lacks intuition and would benefit from a visualization.

**Questions For Authors:**

1. How would the pipeline of eps-VAE and LDM benefit from even higher downsampling factors?
  - Could the efficiency of the diffusion model be improved (both in terms of training and sampling) without hurting the generation quality significantly?
2. Why does the reconstruction performance degrade when increasing the number of steps starting from 3?

Convincing responses to these questions could alleviate my concerns regarding the limitations of the proposed approach.

**Relation To Broader Scientific Literature:**

The paper is using a lot of ideas from recent improvements of denoising generative models like the rectified flow schedule with velocity parameterization, the VAE encoder architecture from StableDiffusion / Latent Diffusion Models, a similar combination of loss functions for training with reconstruction, LPIPS, and adversarial loss terms (but adjusted for denoising), denoiser architectures from ADM and DiT, a logit-normal distribution for timestep sampling during training from StableDiffusion 3, but it explores the best configuration for a different setting being: Lightweight generative image compression instead of training the second-stage LDM, e.g, for text-to-image generation.
For this particular task, certain design questions have to be answered differently showed by the advantage of the UNet-based ADM architecture over a Transformer-based DiT, for example.
There is a relevant related work [6] that proposes a similar two(/ three with VAE)-stage approach with two diffusion models but at a higher level of (semantic) compression, still using the standard VAE for the first image compression.

There has not been that much prior work focusing on improving the VAE / tokenizer part of latent diffusion or autoregressive approaches, but a lot of concurrent work [1, 2, 3, 4]. Prior works mostly used existing pre-trained VAEs from StableDiffusion or trained a similar architecture with different hyperparameters and downsampling factors like MAR [5] for example.

- [1] Deep Compression Autoencoder for Efficient High-Resolution Diffusion Models. ICLR 2025
- [2] Reconstruction vs. Generation: Taming Optimization Dilemma in Latent Diffusion Models. arxiv 2025
- [3] Exploring Representation-Aligned Latent Space for Better Generation. arxiv 2025
- [4] FlexTok: Resampling Images into 1D Token Sequences of Flexible Length. arxiv 2025
- [5] Autoregressive Image Generation without Vector Quantization. NeurIPS 2024
- [6] Würstchen: An efficient architecture for large-scale text-to-image diffusion models. ICLR 2024

**Theoretical Claims:**

There are no theoretical claims that require proofs.

---

> ### Author Rebuttal · Authors · 2025-04-01
>
> Thank you for your constructive comments. We will rephrase our claims and reduce the redundancy as suggested, and include suggested related work in the revision. Below we provide a point-by-point response to all of your questions. Please let us know if you have any further questions.
>
> **Q1: Because of the denoising procedure with 3 network forward evaluations used in the paper, the decoding is more expensive than with a standard VAE, which the paper misses to openly discuss. This limits its application in cases with real-time visualization, e.g., during the generation process of images.**
>
> Thank you for pointing out this limitation, and we will include a detailed discussion of it in the revised version. However, we would like to note that our denoising process demonstrates promising results even with a single iteration (please refer to Figure 3, left and right, in the original paper). Consequently, this allows our model to be adapted for scenarios with latency-sensitive requirements, such as real-time visualization during image generation as mentioned, by reducing the decoding step to a single pass.
>
> **Q2: How would the pipeline of eps-VAE and LDM benefit from even higher downsampling factors? Could the efficiency of the diffusion model be improved (both in terms of training and sampling) without hurting the generation quality significantly?**
>
> To address the questions regarding higher downsampling factors, we present additional results for a 32 x 32 downsampling factor in the table below, comparing them to our 16 x 16 results. Notably, Epsilon-VAE achieves a 25% improvement in generation quality over SD-VAE at the 32 x 32 factor, alongside a 3.2x inference speedup than SD-VAE at the 16 x 16 factor with comparable FID (highlighted in bold). We observed similar training speedups for latent diffusion models utilizing Epsilon-VAE at this higher downsampling rate. These gains are more pronounced than those observed when increasing the downsampling factor from 8 x 8 to 16 x 16 (where we observe 22% quality improvement and 2.3x inference speedup, respectively, as shown in the original paper). These findings strongly suggest that the benefits of the Epsilon-VAE and LDM pipeline are amplified with higher downsampling factors. We will include detailed results in the revised paper.
>
> | Method | Downsample factor | Throughput | rFID | FID w/o CFG |
> | -------- | :-------: | :-------: | :-------: | :-------: |
> | **SD-VAE** | **16 x 16** | **1220** | **2.93** | **14.59** |
> | SD-VAE | 32 x 32 | 3991 | 9.33 | 21.31 |
> | Epsilon-VAE (M) | 16 x 16 | 1192 | 1.91 | 10.68 |
> | **Epsilon-VAE (M)** | **32 x 32** | **3865** | **3.80** | **15.98** |
>
> **Q3: Why does the reconstruction performance degrade when increasing the number of steps starting from 3?**
>
> To enable large step sizes for the reverse process during inference, we introduced the denoising trajectory matching loss to implicitly model the conditional distribution $p(x_0|x_t)$, shifting the denoising distributions from traditional Gaussian to non-Gaussian multimodal forms (please refer to [1*] for a detailed discussion on this). However, the assumptions underpinning this approach are most effective when the total number of denoising steps remains small. Consequently, there appears to be an optimal range or "sweet spot" for the number of total inference steps. We will elaborate on this phenomenon and provide further clarification in the revised version.
>
> [1*] Tackling the Generative Learning Trilemma with Denoising Diffusion GANs, ICLR 2022.
>
> **Q4: The qualitative comparisons in the paper and the appendix are not very convincing showing only very minor differences in high-frequency details.**
>
> This is because in the original paper, we show compressed high-resolution images with Epsilon-VAEs using low downsampling factors. We will include additional uncompressed visual results under more changeling settings (e.g., 128 x 128 images with 16 or 32 downsampling factors) in the revised version to highlight our advantages.
>
> **Q5: The proposed reversed logarithm mapping for the timesteps during sampling lacks intuition and would benefit from a visualization.**
>
> As detailed in our response to Q3, the denoising trajectory matching loss results in denoising distributions that deviate from the standard Gaussian. This deviation suggests that the conventional uniform sampling approach may no longer be optimal. Hence, we empirically investigated alternative sampling strategies and found the reversed logarithm mapping to yield the best performance. We will clarify the intuition and provide additional visualizations as suggested in the revision.

---

> > ### Comment · Reviewer_3ACU · 2025-04-03
> >
> > I appreciate the rebuttal from the authors.
> > Regarding related work, I found another similar paper [1] that does not impact the significance of the contributions but should be discussed in the final version. While following a different motivation, this paper essentially also proposes a VAE with a denoising decoder but uses the encoding as the "initial noise" instead of as conditioning for a standard diffusion model starting from a standard Gaussian distribution. This could also be an interesting idea for speeding up the proposed approach.
> >
> > [1] Minimizing Trajectory Curvature of ODE-based Generative Models. ICML 2023
> >
> > Regarding the rebuttal, I would like to first inform the authors (in case they have not noticed) that it is possible to include anonymous links to additional figures in the rebuttal, which I think would be beneficial to address some of my concerns but also from other reviewers, e.g., regarding the provided qualitative results in the paper with unclear zoom-in boxes due to image compression (cf. reviewer NjT6).
> > Furthermore, I would like to comment on some of the points regarding my review:
> >
> > > Q1 Multi-step denoising / decoding
> >
> > Thank you for pointing me to Fig.3. However, this figure does not contain a comparison with the baseline SD-VAE but only shows an ablation study. I still think this fair comparison in terms of NFEs would be important.
> >
> > > Q2  Higher compression rates
> >
> > Thank you very much for providing additional experimental results. I think these results are convincing and therefore this concern has been addressed satisfactorily.
> >
> > > Q3 Degraded reconstruction performance with more than 3 steps
> >
> > The response makes sense, thanks!
> >
> > > Q4 Unconvincing qualitative comparisons
> >
> > If you already have qualitative results for the revised version, it would be helpful to provide them for the rebuttal via *anonymous* links. Otherwise, I cannot evaluate whether these visual results are convincing or not.
> >
> > > Q5 Intuition and visualization of reversed logarithm mapping
> >
> > Again, a visualization would have been great to have for the rebuttal.
> >
> > The rebuttal addresses some but not all of my concerns. After reading all reviews, I am still leaning towards accepting this paper, but currently slightly more towards weak accept than accept because of the unresolved concerns.

---

> > > ### Author Response · Authors · 2025-04-05
> > >
> > > We thank the reviewer for the valuable feedback and for bringing reference [1] to our attention. We agree that this is indeed relevant and appreciate the suggestion. We will incorporate a discussion of [1] in the revised version. We also find the idea of potentially leveraging its approach to speed up our model interesting and will note this as a direction for future work.
> > >
> > > Below we provide point-by-point responses to your comments. We hope they could address your remaining concerns and help with finalizing the final rating.
> > >
> > > **Q1 Multi-step denoising / decoding.**
> > >
> > > We provide a direct comparison between SD-VAE and our *one-step* Epsilon-VAE in the table below. The table presents image reconstruction quality on ImageNet 256 x 256 with the 8 x 8 downsampling factor. We include two variants: Epsilon-VAE (B), which has a similar inference speed to SD-VAE, and Epsilon-VAE (M), which matches SD-VAE in the number of parameters. As shown, both Epsilon-VAE (B) and (M) outperform SD-VAE across all metrics. These results confirm the effectiveness and efficiency of our one-step models compared to SD-VAE.
> > >
> > > | Method | rFID | PSNR | SSIM |
> > > | -------- | :-------: | :-------: | :-------: |
> > > | SD-VAE | 0.74 | 25.68 | 0.820 |
> > > | Epsilon-VAE (B) | 0.57 | 25.91 | 0.826 |
> > > | Epsilon-VAE (M) | 0.51 | 26.45 | 0.830 |
> > >
> > > **Q4 Unconvincing qualitative comparisons.**
> > >
> > > Please find the qualitative results with different downsampling ratios (including 8 x 8, 16 x 16, and 32 x 32) on ImageNet 128 x 128 in https://anonymous.4open.science/r/ICML2025-4F8E/ (the file name is "3ACU-Q4_qualitative.png", best viewed when zoomed-in). We use these settings for visualization to highlight the differences since they are more challenging for reconstruction. Epsilon-VAE achieves higher fidelity and better perceptual quality, especially under extreme downsampling factors.
> > >
> > > **Q5 Intuition and visualization of reversed logarithm mapping.**
> > >
> > > Please find the visualization results of both uniform sampling and reversed logarithm sampling (including 1-step, 2-step, and 3-step models) on ImageNet 128 x 128 in https://anonymous.4open.science/r/ICML2025-4F8E/ (the file name is "3ACU-Q5_sampling.png", best viewed when zoomed-in). We can find that the reversed logarithm sampling method could lead to improved details in local regions with complex textures or structures, especially when the number of sampling steps is increased to three.

---

### Official Review · Reviewer_FhpZ · 2025-03-11

**Overall Recommendation:** 3

**Summary:**

In this work, the authors propose using denoising diffusion model as the decoder in autoencoder for image reconstruction and generation. $\epsilon$-VAE develops denoising decoder conditioned on the learnable latents. The work includes solid experiments in validating the design choices for image reconstruction. Further DiT-based latent diffusion models on $\epsilon$-VAE latents achieves comparable performance as popular SD-VAE.

**Claims And Evidence:**

The authors claim superior performance on image reconstruction which is well supported by the experimental results. But I find the generation performance of $\epsilon$-VAE latents not fully examined. In particular, in Table 3, the authors shows better FID than DiT with SD-VAE under the diffusion setting reported in this work. However, the reported performance is worse than numbers reported in the original DiT paper. Also, the authors only report results without CFG. Whereas CFG is broadly applied to diffusion models and should be compared to fully validate the performance of proposed $\epsilon$-VAE.

**Essential References Not Discussed:**

N/A

**Experimental Designs Or Analyses:**

Experimental designs and analysis are valid.

**Methods And Evaluation Criteria:**

Yes, evaluation and benchmark make sense.

**Other Comments Or Suggestions:**

N/A

**Other Strengths And Weaknesses:**

N/A

**Questions For Authors:**

1. My main question is the performance of $\epsilon$-VAE on image generation as mentioned in "Claims And Evidence". It would greatly help validate the contribution of the work by showing how $\epsilon$-VAE works with CFG.
2. In latent diffusion, DiT with $\epsilon$-VAE shows better performance than SD-VAE under the setting of this work. Is the setting optimized for $\epsilon$-VAE or is it also benefitting standard DiT with SD-VAE?
3. Can authors provide more details of how inference is conducted on higher resolutions with models trained at 128 x 128 images?

**Relation To Broader Scientific Literature:**

The paper is very related to the previous latent diffusion models which relies on autoencoder to acquire latent space. The work follows the pipeline and aims at improving the autoencoder design.

**Theoretical Claims:**

Theoretical claims in the work are valid.

---

> ### Author Rebuttal · Authors · 2025-04-01
>
> Thank you for your valuable comments. Below we provide a point-by-point response to all of your questions. Please let us know if you have any further questions.
>
> **Q1: The reported performance in Table 3 is worse than numbers reported in the original DiT paper.**
>
> We did the following modifications to the original DiT training recipe as mentioned in Section C of the original paper, leading to a slight drop in performance: (1) we reimplement the pipeline with JAX for training models on TPUs; (2) for simplicity and training stability, we remove the variational lower bound loss term; (3) we reduce the number of training steps to 1M to conserve compute.
>
> **Q2: It would greatly help validate the contribution of the work by showing how ϵ-VAE works with CFG.**
>
> We provided results with CFG under the 8 x 8 downsample factor in the table below, where we find that Epsilon-VAE (M) performs relatively 20% better than SD-VAE and further improvements are obtained after we scale up our model to Epsilon-VAE (H). These results are consistent with the results without CFG, confirming the effectiveness of our model. We will provide more detailed results of other models and under different downsample factors in the revised version.
>
> | Method | FID w/ CFG |
> | -------- | :-------: |
> | SD-VAE | 3.51 |
> | Epsilon-VAE (M) | 2.83 |
> | Epsilon-VAE (H) | 2.69 |
>
> **Q3: In latent diffusion, is the DiT setting optimized for Epsilon-VAE or is it also benefitting standard DiT with SD-VAE?**
>
> Our training protocol for latent diffusion was consistently applied across all evaluated VAE models. The three minor modifications to the original DiT training recipe, as outlined in our response to Q1, were implemented solely to conserve computational resources. Our current setting does not favor either our Epsilon-VAE or the standard SD-VAE, but it ensures  a fair basis for comparison.
>
> **Q4: Can authors provide more details of how inference is conducted on higher resolutions with models trained at 128 x 128 images?**
>
> Inference on higher resolutions works out-of-the-box. Since our model is a fully convolutional UNet, it can directly process images with resolutions exceeding the 128 x 128 training size without any modifications to the input.

---

### Official Review · Reviewer_RVFr · 2025-03-13

**Overall Recommendation:** 2

**Summary:**

The author propose a new autoencoder paradigm, in which decoder is in the form of a diffusion model. The advantage of this is that it has better reconstruction quality compared with standard VAE. The proposed architecture is straight-forward, by directly upsampling encoded latents and then run diffusion at the input resolution to get the original signal backs. They introduce adversarial loss to encourage small reconstruction error from the diffusion decoder. In their main experiment, they use existing architecture, such as VQGAN encoder and discriminator for this purpose. They demonstrate epsilon-VAE achieve better reconstruction than other latent space VAE.

**Claims And Evidence:**

Yes.

**Essential References Not Discussed:**

They have good coverage.

**Experimental Designs Or Analyses:**

The experiment designs make sense. However, I would argue a benchmark on computation at various decoding resolution is necessary. The memory could be quadratically increased with the input size, which is not desirable.

**Methods And Evaluation Criteria:**

Method-wise, I am not convinced about the epsilon-VAE idea. At a high-level, they are proposing a direction opposite to latent diffusion. Instead of doing diffusion at latent space (which was first proposed in Stable Diffusion for budge reduction), it does that instead in the original input space. It is not suprising that they can achieve better reconstruction performance, (similar argument is already proven by Imagen and latter work) but the computation is much higher. Besides, it is counter-intuitive to replace a single step decoding with a iterative process, which is more computationally heavy.

Evaluation criteria makes sense.

**Other Comments Or Suggestions:**

none

**Other Strengths And Weaknesses:**

Strength:
1. Overall the authors did a great job for experiment design which has a thorough coverage of their design choices and potential use cases, e.g., image-conditioned diffusion training.

2. The proposed framework is generic, which works with existing off-the-shelf encoder network and diffusion architecutres.

Weakness:
1. As is stated above, my major concern is on its high-level insights. They are proposing an interesting direction opposite to latent diffusion. Instead of doing diffusion at latent space (which was first proposed in Stable Diffusion for budge reduction), it does that instead in the original input space. It is not suprising that they can achieve better reconstruction performance, (similar argument is already proven by Imagen and latter work) but the computation is much higher. Experiment evidences wise, it does not justify why such a design choice is worthy to pursuit in my opinion. Besides, it is counter-intuitive to replace a single step decoding with a iterative process, which is more computationally heavy.

**Questions For Authors:**

none.

**Relation To Broader Scientific Literature:**

Many generative models (e.g., image, video, and other LMM) use latent diffusion style to train. The proposed framework could speed up the training by combining the autoencoding and generation step together.

**Theoretical Claims:**

no theoretical claims.

---

> ### Author Rebuttal · Authors · 2025-04-01
>
> Thanks for your valuable comments. Below we provide a point-by-point response to your questions. Please let us know if you have any further questions.
>
> **Q1: Clarification on high-level insights.**
>
> Thank you for raising this concern. Rather than opposing latent diffusion, we offer a complementary perspective by demonstrating that the diffusion process can also be applied to the outer structure of the latent diffusion model – the autoencoder part. Our goal is to revisit the standard autoencoder framework and integrate diffusion into the decoding step to enhance the entire model’s generative capacity. We empirically show that a latent diffusion DiT integrated with Epsilon-VAE significantly outperforms its SD-VAE counterpart, with the performance gap being significant in high compression regime.
>
> **Q2: Diffusion-based iterative decoding.**
>
> While replacing single-step decoding with an iterative process may seem counter-intuitive due to increased computational cost, the diffusion-based decoder addresses this concern in three key ways. First, it offers scalable inference, where even a single-step variant already outperforms a plain VAE decoder, and additional steps further enhance quality (please refer to Figure 3, left and right, and Table 7 in the original paper, where a single-step Epsilon-VAE-lite achieves 7.18 rFID and a VAE obtains 11.15 rFID in the same setting).  Second, it provides controllable trade-offs between computation and visual fidelity, allowing the number of steps to be adjusted at inference time based on application needs. Third, as shown in Section 4.2 of the main paper and our response to Q3 below, it enables training under higher compression ratios, which helps offset the added cost of iterative decoding by reducing the size of latent representations.
>
> An additional advantage of scaling the autoencoder over the latent model lies in computational efficiency. Recent trends show latent diffusion models increasingly adopt Transformer architectures (e.g., DiT), where self-attention scales quadratically with input resolution. In contrast, our convolution-based UNet decoder offers more favorable linear scaling. *As models grow, shifting complexity to the autoencoder helps reduce the burden on the latent model, leading to a more efficient overall system.*
>
> **Q3: A benchmark on computation at various decoding resolutions.**
>
> In response to the reviewer's valuable point regarding computational cost at various decoding resolutions, we conducted experiments with Epsilon-VAE using a 16 x 16 downsampling factor. Our analysis reveals that: (1) the dominant factor in the overall memory footprint during the generation process is the LDM; (2) increasing the decoding resolution leads to a more substantial increase in the memory requirements of the LDM compared to the Epsilon-VAE decoder itself; (3) Epsilon-VAE outperforms SD-VAE under the same 8 x 8 downsampling factor at the 256 x 256 resolution, but with slightly worse throughput and memory efficiency; and (4) importantly, by employing a 16 x 16 downsampling factor instead of 8 x 8, Epsilon-VAE demonstrates a significant 2.3x inference speedup and a 3.3x reduction in total memory cost compared to SD-VAE, while maintaining comparable image generation performance (highlighted in bold). This demonstrates the efficiency gains achieved with our proposed approach in mitigating the potential for undesirable decoding memory scaling.
>
> | Method | Resolution | Downsample factor | Throughput | LDM memory (GB) | Decoder memory (GB) | FID w/o CFG |
> | -------- | :-------: | :-------: | :-------: | :-------: | :-------: | :-------: |
> | **SD-VAE** | **256 x 256** | **8 x 8** | **522** | **78.8** | **1.7** | **11.63** |
> | Epsilon-VAE (M) | 256 x 256 | 8 x 8 | 491 | 78.8 | 2.8 | 9.39 |
> | Epsilon-VAE (M) | 128 x 128 | 16 x 16 | 3910 | 5.9  | 1.6 | 11.14 |
> | **Epsilon-VAE (M)** | **256 x 256** | **16 x 16** | **1192** | **20.7** | **3.7** | **10.68** |
> | Epsilon-VAE (M) | 512 x 512 | 16 x 16 | 240 | 82.6 | 6.9 | 9.20 |

---

### Official Review · Reviewer_NjT6 · 2025-03-14

**Overall Recommendation:** 4

**Summary:**

This paper proposes to use a diffusion decoder in an autoencoder training for image generation. The autoencoder is trained with a diffusion loss, together with a LPIPS loss and a GAN loss defined on the one-step generation. The authors show that the proposed method outperforms prior state-of-the-art autoencoders for both reconstruction and image generation.

**Claims And Evidence:**

The key claim of this paper is that a diffusion-loss based autoencoder (with LPIPS and GAN losses) can outperform the prior autoencoders while being efficient efficient. The authors evaluate rFID, PSNR, SSIM for reconstruction, and generation FID, the results show that the proposed method outperforms prior works. Visualization is also consistent with the claim.

**Essential References Not Discussed:**

I did not find key missing related works.

**Experimental Designs Or Analyses:**

The experiments are solid. Authors evaluate rFID, PSNR, SSIM, gFID and show visual samples for comparisons with prior methods for both reconstruction and generation. The number of parameters is also listed in supplementary material (Table 7) and shows a fair comparison. In ablation study, Table 4 shows the effect of different design choices which help understanding the importance of different components for the method.

**Methods And Evaluation Criteria:**

Diffusion models are scalable probabilistic models. Using diffusion loss is thus a promising way to better model the probabilistic decoding and for learning image tokens. Evaluation metrics include rFID, PSNR, SSIM, and generation FID, which are standard in this field. Ideally human evaluation can be added, while it may not be necessary given the large amount of visual samples.

**Other Comments Or Suggestions:**

The zoom-in boxes in Fig. 4 seems to be not very clear even for the GT.

**Other Strengths And Weaknesses:**

See prior sections for strengths. Using diffusion loss for autoencoder training is an important and promising direction to explore, and this work is one of the first in this direction that shows solid improvement on common benchmark ImageNet which are general images. Autoencoders are of key importance for generative models, this work proposes a novel autoencoder with state-of-the-art performance, therefore it is an important contribution to the field.

Minor weakness: The LPIPS and GAN loss are applied on the estimated one-step sample, which may not be accurate and may cause objective bias in theory. Although this can potentially be addressed with finetuning a diffusion decoder with frozen z.

**Questions For Authors:**

Besides rFID, how much improvement it gets from the additional LPIPS and GAN losses for the actual visual quliaty in images?

**Relation To Broader Scientific Literature:**

This work is one of the first works that show using the diffusion loss for the decoder can help training the autoencoders for reconstruction and generation on latent. The proposed method also serves as a novel autoencoder that achieves state-of-the-art quality while being efficient.

**Theoretical Claims:**

Theoretical proofs are not the focus of this work. The formulations are correct.

---

> ### Author Rebuttal · Authors · 2025-04-01
>
> Thanks for your positive feedback. Below we provide a point-by-point response to your questions. Please let us know if you have any further questions.
>
> **Q1: The LPIPS and GAN loss are applied on the estimated one-step sample, which may not be accurate and may cause objective bias in theory. Although this can potentially be addressed with finetuning a diffusion decoder with frozen $z$.**
>
> We acknowledge the reviewer's point regarding potential objective bias due to applying the LPIPS and GAN loss on the estimated one-step sample. However, we would like to emphasize that our Epsilon-VAE differs significantly from traditional diffusion models in that its diffusion decoder is conditioned on encoded latents. This conditioning provides a strong prior about the input image to reconstruct, resulting in a more accurate estimated one-step sample than in typical diffusion scenarios. Therefore, we believe the potential for objective bias is considerably reduced. We agree that finetuning the diffusion decoder with frozen $z$ is a promising avenue for further improvement and will explore this in future work.
>
> **Q2: Besides rFID, how much improvement it gets from the additional LPIPS and GAN losses for the actual visual quality in images?**
>
> We find that the LPIPS loss enhances textural fidelity and structural coherence in generated images, while the GAN loss contributes to sharper high-frequency details. These observations align with their established roles in traditional VAEs. Detailed visual comparisons demonstrating these improvements will be included in the revised version.
>
> **Q3: The zoom-in boxes in Fig. 4 seem to be not very clear even for the GT.**
>
> Thanks for pointing out this. This is because the compressed images were included in the original paper to ensure a reasonable file size. We will provide the original, uncompressed images in the revised version.

---

### Decision · Program_Chairs · 2025-05-01

**Decision:**

Accept (poster)

**Comment:**

The paper proposes a new autoencoder paradigm where the decoder is a diffusion model. The paper shows that this setup achieves better reconstruction quality than standard VAEs. The adversarial loss introduced reconstructs a small error from the diffusion encoder. The paper is deemed interesting for proposing a diffusion-based decoder in a traditional VAE setup. The experiments are solid and extensive.

The main concern that remains seems to stem from the paper exploring the diffusion process in the decoding step, which is more expensive than other options. Due to the nature of research, I don't consider the slower and more expensive method a problem since it explores new avenues. The other concerns raised by the reviewers were addressed in the rebuttal. While there were some minor concerns regarding the lack of human evaluation and potential biases in the loss measures, the majority of the reviewers agreed that the paper is interesting.

Given the positive evaluation, I recommend the paper for acceptance.